# Growth cone-localized microtubule organizing center establishes microtubule orientation in dendrites

Xing Liang[1,2†], Marcela Kokes[1,2†], Richard D Fetter[2], Maria Danielle Sallee[1], Adrian W Moore[3], Jessica L Feldman[1]*, Kang Shen[1,2]*

[1]Department of Biology, Stanford University, Stanford, United States; [2]Howard Hughes Medical Institute, Stanford University, Stanford, United States; [3]RIKEN Center for Brain Science, Wako, Japan

**Abstract** A polarized arrangement of neuronal microtubule arrays is the foundation of membrane trafficking and subcellular compartmentalization. Conserved among both invertebrates and vertebrates, axons contain exclusively 'plus-end-out' microtubules while dendrites contain a high percentage of 'minus-end-out' microtubules, the origins of which have been a mystery. Here we show that in *Caenorhabditis elegans* the dendritic growth cone contains a non-centrosomal microtubule organizing center (MTOC), which generates minus-end-out microtubules along outgrowing dendrites and plus-end-out microtubules in the growth cone. RAB-11-positive endosomes accumulate in this region and co-migrate with the microtubule nucleation complex γ-TuRC. The MTOC tracks the extending growth cone by kinesin-1/UNC-116-mediated endosome movements on distal plus-end-out microtubules and dynein clusters this advancing MTOC. Critically, perturbation of the function or localization of the MTOC causes reversed microtubule polarity in dendrites. These findings unveil the endosome-localized dendritic MTOC as a critical organelle for establishing axon-dendrite polarity.

**\*For correspondence:**
feldmanj@stanford.edu (JLF);
kangshen@stanford.edu (KS)

[†]These authors contributed equally to this work

## Introduction

The ability of our nervous system to function rests on polarized transport within the axons and dendrites of neurons, a task performed by molecular motors running on a polarized microtubule (MT) network. MTs are dynamic cellular polymers that are critical for this transport as well as cell polarity, and division. These essential cellular functions require spatial MT organization with a defined orientation of MT minus and plus ends relative to cellular coordinates. For example, in mature neurons, MTs are organized along the length of neurites with specific orientation; axons contain exclusively plus-end-out MTs, while dendrites have predominantly minus-end-out or both plus-end-out and minus-end-out MTs in invertebrate and vertebrate species, respectively (*Baas et al., 1988*; *Stepanova et al., 2003*; *Stone et al., 2008*). This specific MT organization is critical for normal intracellular vesicular trafficking as different cargoes engage either plus-end-directed or minus-end-directed motor proteins. Dendrites with reversed MT orientation lose dendritic proteins and ectopically accumulate synaptic vesicles (*Maniar et al., 2012*; *Yan et al., 2013*). Despite the importance of proper MT organization to neuronal function, how MTs initially adopt these patterns during neuronal development is largely unknown.

In cells, MT patterning is conferred by microtubule organizing centers (MTOCs) (*Pickett-Heaps, 1969*), cellular sites which nucleate, anchor, and stabilize MT minus ends (*Lüders and Stearns, 2007*; *Paz and Lüders, 2018*; *Sanchez and Feldman, 2017*). During cell division, the centrosome acts as the MTOC to build the mitotic spindle. In many postmitotic differentiated cells such as neurons, centrosomes are inactive as MTOCs and often eliminated, and MTs instead associate

with non-centrosomal sites such as the apical surface of epithelial cells and the nuclear envelope in skeletal muscle (*Nguyen et al., 2011*; *Stiess et al., 2010*). Conceptually, neurons could also use non-centrosomal MTOCs to locally generate compartment-specific organization (*Lüders and Stearns, 2007*; *Paz and Lüders, 2018*; *Sanchez and Feldman, 2017*); however, the identity of these MTOCs or their molecular composition is currently unknown.

While it is perhaps easier to imagine how plus-end-out MTs in neurites might arise as an extension of the plus-end-cortical MT arrangement during cell division, the origin of a minus-end-out MT arrangement is harder to grasp since the minus ends point away from the cell body, suggesting a distinct mechanism. Several mechanisms have been proposed to contribute to minus-end-out MT bias in mature dendrites, including Golgi outpost-mediated MT nucleation, MT nucleation at branch sites, MT nucleation from the base of cilia, MT sliding or steering, and minus-end growth. In mature *Drosophila* sensory neurons, Golgi outposts are frequently located at dendritic branch sites and can nucleate minus-end-out MTs (*Ori-McKenney et al., 2012*). However, the role of Golgi outposts during development is unclear, and one study showed that removal of Golgi outposts from dendrites does not alter minus-end-out MTs (*Nguyen et al., 2014*). Furthermore, a recent study found that a subpopulation of early endosomes at branch sites house canonical Wnt signaling proteins and nucleate MTs to influence dendritic MT polarity after branching has occured (*Weiner et al., 2020*). In the dendrites of *C. elegans* ciliated neurons, microtubules are thought to be nucleated from the base of the cilium, although this mechanism by definition is absent from non-ciliated neurons (*Harterink et al., 2018*). In rodent primary hippocampal cultures, short MTs could be delivered from the cell body to neurites by molecular motor-based sliding on other MTs (*Rao and Baas, 2018*). Indeed, kinesin-1, a plus-end-directed motor is required for the minus-end-out MTs in the mature dendrite of a *C. elegans* motor neuron (*Yan et al., 2013*). However, direct observations of MT sliding in dendrites in vivo have not been reported. Additionally, kinesin-2 is required for minus-end-out MTs in *Drosophila* sensory dendrites, although through an alternate proposed mechanism of steering growing MT plus ends along pre-existing MTs (*Weiner et al., 2016*). Surprisingly, MT minus ends themselves can grow in distal dendrites to form a uniformly minus-end-out MT array in branched *Drosophila* sensory dendrites. Loss of the MT minus end stabilizing protein Patronin reduces minus-end-out MTs in terminal dendrite branches, while Patronin overexpression increases the percentage of minus-end-out MTs (*Feng et al., 2019*). While it is clear that minus end MT growth promotes minus-end-out MT bias in terminal branches, this mechanism is not required for establishing a population of minus-end-out MTs within the primary dendrites from which the terminal branches grow. Thus, although many mechanisms have been observed and proposed to maintain dendrite MT polarity, it remains unknown what the MT polarity is during dendrite outgrowth and what mechanisms generate MT polarity during development.

Here, we followed dendritic MT organization and orientation in a single highly branched *C. elegans* sensory neuron from its birth. The primary anterior dendrite of PVD contains predominantly minus-end-out MTs (*Taylor et al., 2015*), and we found that dendritic MTs were organized by a striking MTOC localized to the dendritic growth cone. Due to its unique subcellular position, this dendritic growth cone MTOC (dgMTOC) generates a minus-end-out MT array toward the cell body that populates the length of the anterior dendrite and a short plus-end-out MT array reaching toward the growing dendrite tip. The dgMTOC is highly mobile and continuously tracks with the dendritic growth cone as it advances through the surrounding tissue. dgMTOC localization to the dendritic growth cone is achieved through the balanced action of motor proteins; kinesin-1 transports the dgMTOC distally on plus-end-out MTs, while dynein opposes this movement to prevent dgMTOC dispersal. Loss or ectopic localization of the dgMTOC leads to an absence of minus-end-out MTs in the PVD dendrite, indicating it is required for the initial establishment of dendritic minus-end-out MTs. Finally, we found that dgMTOC activity colocalizes and co-traffics with RAB-11 endosomes, suggesting a structural basis for dgMTOC function. Together, these results identify a local dendritic MTOC that first establishes the polarized array of MTs necessary for neuronal function.

## Results

### An active MTOC localizes to the growth cone of the outgrowing primary dendrite

We explored the origins of MT organization using the PVD neuron in *C. elegans*, a sensory neuron with stereotypical morphology of its axon and non-ciliated dendrites (*Albeg et al., 2011*). Early PVD morphogenesis is temporally and spatially stereotyped: the axon always grows out first, followed by the anterior and then posterior dendrite, with all emerging neurites oriented in the direction of their mature neurite pattern. The elaborate branches develop after the primary dendrites fully extend (*Figure 1A*, *Figure 1—figure supplement 1A and B*). We previously showed that the mature anterior primary dendrite contains largely minus-end-out MTs while the posterior dendrite and axon have plus-end-out MTs (*Taylor et al., 2015*). To investigate the establishment of minus-end-out MTs during dendrite outgrowth (step III in *Figure 1A*), we visualized the endogenous localization of the MT plus-end tracking protein EBP-2::GFP/EB1 in developing PVD anterior dendrites and found that numerous EBP-2 comets emerged from a single region within the distal neurite, suggesting the presence of a dendritic growth cone MTOC (dgMTOC, *Figure 1B* and *Video 1*). To characterize the directionality of these EBP-2 comets, we plotted the frequency of plus-end-out and minus-end-out MTs at interval distances in the anterior dendrite, setting the center of the apparent dgMTOC as zero (*Figure 1C*, *Figure 1—source data 1*). Near the most distal region of the dendrite (~3–6 µm from the dendrite tip), almost all the EBP-2 comets move toward the distal tip indicating plus-end-out microtubules, while nearly all the comets along the shaft of the anterior dendrite move toward the cell body indicating only minus-end-out MTs in this region (*Figure 1B and C*). The emergence of a high frequency of EBP-2 comets from a discrete region near the dendrite tip (mean = 0.44 comets/s±0.11 (SD) in a 3.2 µm region), which also encompasses a transition zone of MT directionality, is consistent with the presence of a local MTOC (*Figure 1B and C*). This MT organization is apparent as soon as a morphologically distinct growth cone-tipped anterior dendrite has outgrown from the cell body and remains throughout the outgrowth of the anterior dendrite (data not shown). In addition, the number of comets generated near the dendrite tip greatly exceeds comets from any other discrete point or region in the dendrite or from the cell body throughout the morphogenesis of the anterior dendrite, indicating that the dgMTOC is the only apparent MTOC. The axon and posterior dendrite, which contain predominantly plus-end-out MTs, do not contain similar structures (*Figure 1B* and *Figure 1—figure supplement 1C*). Significantly, we find evidence for a similar dgMTOC near the tip of outgrowing *Drosophila melanogaster* Class I sensory vpda primary dendrites (*Figure 1—figure supplement 1D* and *Video 2*), suggesting a conserved mechanism for the genesis of minus-end-out MTs in the primary dendrite.

To further confirm the existence and location of the MTOC, we performed time-lapse imaging on worms expressing GFP::TBA-1/α-tubulin in PVD, which allowed us to track MT polymerization and depolymerization events. TBA-1 was enriched in a region immediately behind the growth cone (*Figure 1—figure supplement 1E and F*, *Figure 1—source data 1*), consistent with the presence of a MTOC. The relatively low number of MTs in *C. elegans* neurons (*Yogev et al., 2016*) allowed us to identify individual MTs on kymographs. This analysis also revealed an apparent dgMTOC from which plus-end-out MTs extended distally toward the dendritic tip and minus-end-out MTs extended proximally toward the cell body (blue and red lines, respectively, in *Figure 1D*). Consistent with the direction of EBP-2 comets, we observed numerous polymerization events originating from the dgMTOC and directed toward the cell body (red lines in *Figure 1D*). A similar frequency of depolymerization events was also observed in this region (green lines in *Figure 1D* and *Figure 1E*, *Figure 1—source data 1*), indicating that the majority of dgMTOC-derived minus-end-out MTs are highly dynamic. In summary, both EBP-2 and TBA-1 dynamics suggest an active MTOC localizes to the dendritic growth cone region.

### γ-TuRC localizes to the growth cone region during development

Microtubules are nucleated by the conserved γ-tubulin ring complex (γ-TuRC), a ring of γ-tubulin complex proteins (GCPs) that templates the assembly of new MTs (*Oakley et al., 1990*; *Zheng et al., 1995*). γ-TuRCs localize to MTOCs, including the centrosome, the best studied MTOC which generates MTs from within its pericentriolar material (PCM) in dividing animal cells to form the

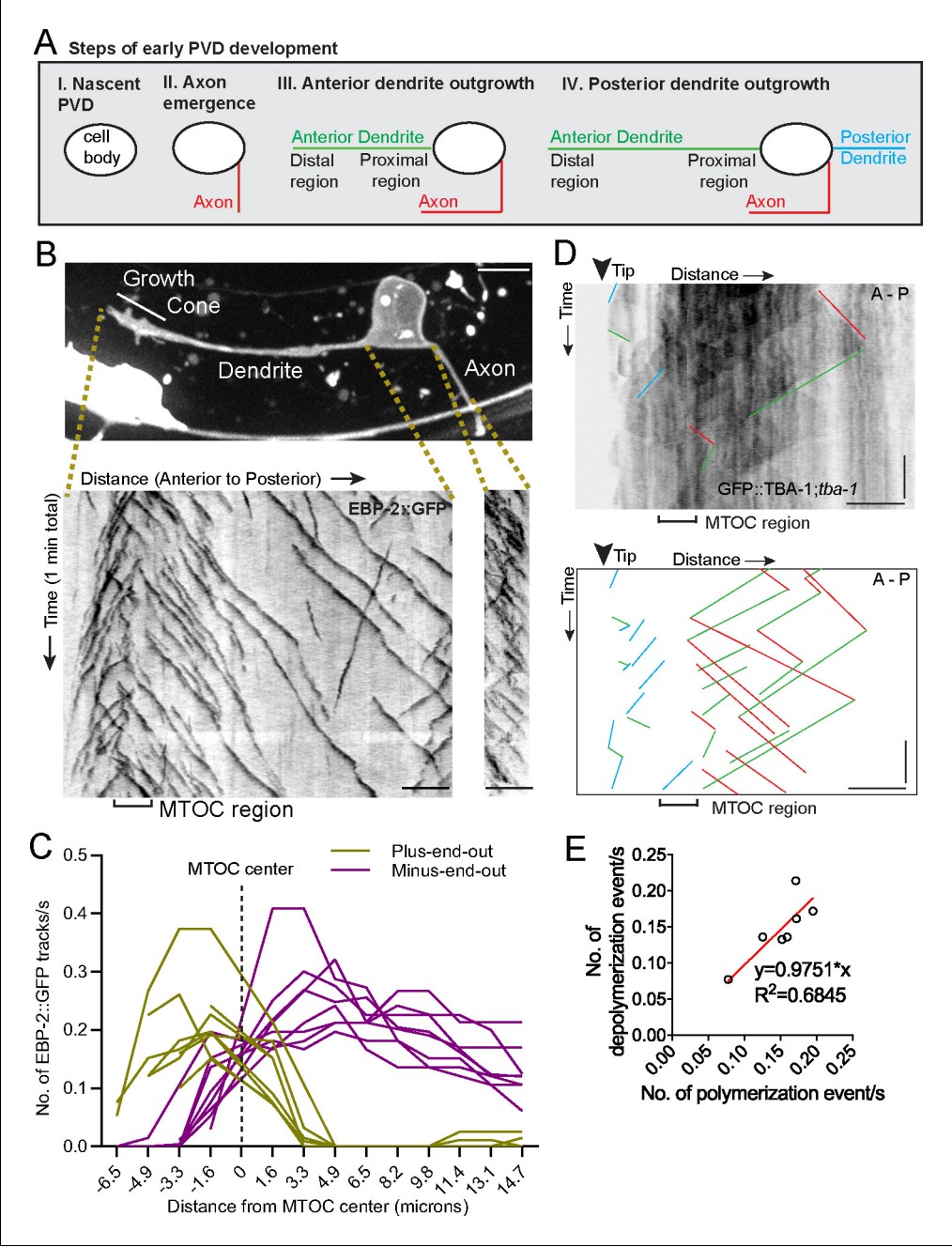

**Figure 1.** An active MTOC localizes to the growth cone of the outgrowing PVD primary dendrite. (**A**) Diagram of the spatiotemporal sequence of PVD neurite emergence and outgrowth during early development. (**B**) Top: A labeled example of PVD morphology during early anterior dendrite outgrowth. Bottom: Kymographs of EBP-2:: GFP in an outgrowing dendrite (left) and axon (right). Scale bar, 5 μm. (**C**) Number of plus-end-out and minus-end-out EBP-2 tracks per second at interval distances from the MTOC center (dashed line) in eight individual animals. (**D**) Kymograph of GFP::TBA-1 in the growth cone region. The GFP::TBA-1 was expressed in a *tba-1* null mutant background to get a better incorporation of the GFP::TBA-1. Blue lines, growing plus-end-out MTs in distal region; red lines, growing minus-end-out MTs in proximal region; green lines, retracting MTs; horizontal scale bar, 5 μm; vertical scale bar, 10 s. (**E**) Frequency of polymerization relative to depolymerization events in the proximal region of the anterior dendrite (n = 7 individual animals). All images are lateral views oriented with anterior to the left and ventral down.

The online version of this article includes the following source data and figure supplement(s) for figure 1:

**Source data 1.** Quantification data for *Figure 1* and *Figure 1—figure supplement 1*.

*Figure 1 continued on next page*

*Figure 1 continued*

**Figure supplement 1.** The dgMTOC is unique to outgrowing anterior dendrites and is conserved in *C. elegans* and *D. melanogaster*.

mitotic spindle (*Conduit et al., 2015*). To understand the molecular components of the dgMTOC, we examined the localization of endogenous GFP::GIP-1/GCP3 and GIP-2::GFP/GCP2 – core components of the γ-TuRC – during dendrite outgrowth. We found that GIP-1 and GIP-2 consistently localized near the dendrite tip in a cluster of punctate structures which track with the growth cone in the outgrowing anterior dendrite (*Figure 2A and B*). GIP-1 precisely localized to the site from which EBP-2 comets originated (*Figure 2C*) and quantification of the distribution and location of GIP-1 and GIP-2 clusters showed that their location closely match the dgMTOC regions defined by EBP-2 comets (*Figure 2D and E*, *Figure 2—source data 1*). These γ-TuRC clusters are specific to the outgrowing anterior dendrite since no GIP-1 or GIP-2 clusters were found in the cell body, axon, or posterior dendrite during their outgrowth (*Figure 2A*, *Figure 2—figure supplement 1A and B*), consistent with the lack of an apparent MTOC in these structures. Furthermore, growth cone-localized GIP-2 was visible as soon as the anterior dendrite emerged (*Figure 2—figure supplement 1C*) but absent near the tip of mature dendrites (*Figure 2—figure supplement 1D*), suggesting that the dgMTOC plays a role in the establishment of MT organization during dendrite development. Significantly, GIP-2 also localized close to the dendrite tip during dendrite outgrowth in the *C. elegans* DA9 motor neuron (*Figure 2—figure supplement 1E*), further supporting a more general role for the dgMTOC in establishing dendritic minus-end-out MTs.

## γ-TuRC is required for minus-end-out MT polarity in the dendrite

To investigate the significance of the dgMTOC and directly test if γ-TuRC is required for the dgMTOC activity, we performed conditional knockdown of GIP-1 in the PVD lineage using the ZIF-1/ZF degradation system previously established in *C. elegans* (*Armenti et al., 2014*; *Sallee et al., 2018*). We inserted a ZF tag into the endogenous *gip-1* locus, enabling the controlled degradation of endogenous GIP-1 upon expression of the E3 ubiquitin ligase substrate-recognition subunit ZIF-1. Expression of ZIF-1 in the developing PVD lineage led to a dramatic reduction of punctate GFP::GIP-1 signal in ~80% of outgrowing PVD anterior dendrites (GIP-1[PVD(-)], *Figure 2—figure supplement 1F*). We then examined EBP-2 comets in GIP-1[PVD(-)] animals, which revealed a striking reversal of anterior dendrite MT polarity leading to plus-end-out orientation in both outgrowing and mature dendrites in about half of the animals (*Figure 2F and G*, *Figure 2—source data 1*). Unlike in wild-type (wt) animals where the majority of EBP-2 comets were generated by the dgMTOC, in GIP-1[PVD(-)] animals, many dendritic comets originated from the cell body (*Figure 2F* and *Video 3*). The incomplete penetrance of this phenotype in GIP-1[PVD(-)] animals is likely due to partial knockdown of GIP-1, as shown by the bimodal distribution of the MT polarity phenotype (*Figure 2G*). These results indicate that γ-TuRC is a critical component of the dgMTOC and that the dgMTOC is essential for establishing minus-end-out MTs in developing and mature dendrites.

## RAB-11 endosomes colocalize with dgMTOC components

To determine the subcellular ultrastructure of the dgMTOC, we performed serial sectioning electron microscopy (EM) reconstruction of developing distal PVD dendrites. We began by reconstructing MTs to identify the dgMTOC

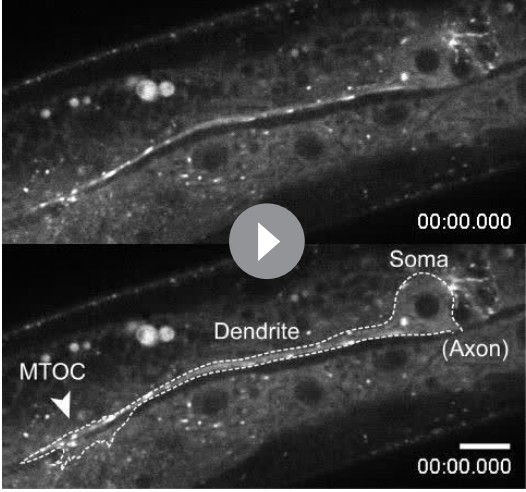

**Video 1.** EBP-2::GFP comets reveal an MTOC in the outgrowing anterior dendrite growth cone of a wt PVD neuron in *C. elegans*.
https://elifesciences.org/articles/56547#video1

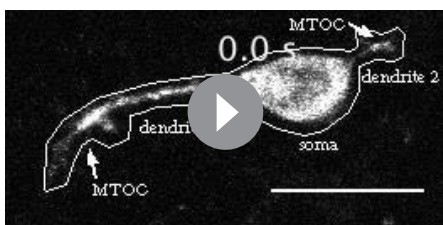

**Video 2.** EB1::GFP comets reveal an MTOC near the outgrowing dendrite tip of vpda in *D. melanogaster*. https://elifesciences.org/articles/56547#video2

region in our sections. In our EM series, numerous short, staggered MTs were observed with their ends unambiguously identified. Consistent with our EBP-2 and TBA-1, the highest MT density was found ~2–4 µm from the dendritic tip, consistent with the presence of an MTOC in this region (*Figure 3A–C*, *Figure 3—source data 1*).

Serial EM reconstruction showed no evidence of a centriole or Golgi outpost in the dgMTOC area. Instead, numerous clear- and dense-core vesicles were found in between the MT arrays (*Figure 3A and B*). In particular, clusters of clear-core vesicles (*Figure 3B*, white) were found in the region with the highest MT numbers suggesting that GIP-1 and GIP-2 might localize to these vesicles to function in the dgMTOC. To investigate the molecular identity of these vesicles, we expressed GFP-tagged markers to label different vesicles and membrane compartments. Synaptic vesicles are the most abundant clear-core vesicles in neurons; however, the synaptic vesicle marker RAB-3, which robustly localized to the axon (*Maniar et al., 2012*), was not enriched in the dendritic growth cone region (*Figure 3—figure supplement 1A*). Consistent with our EM results, the early/medial Golgi cisternae markers RER-1 (*Sato et al., 2011*) and AMAN-2/alpha-mannosidase 2, the latter of which localizes to Golgi outpost MTOCs at mature dendrite branch sites in *D. melanogaster* (*Ori-McKenney et al., 2012*), were only present in the cell body but not in the growth cone (*Figure 3—figure supplement 1B and C*). Surprisingly, the late-Golgi compartment marker RAB-6.2 (*Zhang et al., 2012*) was enriched in the growth cone in a localization pattern similar to that of γ-TuRC (*Figure 3—figure supplement 1B and C*). In addition to localizing to Golgi stacks, RAB-6.2 also plays a role in recycling cargo molecules from endosomes to the trans-Golgi network in neurons (*Zhang et al., 2012*). Therefore, we further examined recycling endosomes and found that the recycling endosome marker RAB-11.1 co-localized and moved together with RAB-6.2-labeled vesicles in the growth cone (*Figure 3—figure supplement 1D and E*). Notably, RAB-11.1 showed striking co-localization with GIP-2 (*Figure 3D*), with 81.7 ± 3.5% (SEM, n = 11) of RAB-11.1 fluorescence overlapping with GIP-2 and 80.6 ± 5.2% (SEM, n = 11) reciprocal GIP-2 fluorescence overlapping with RAB-11.1 in the growth cone (*Figure 3E*, *Figure 3—source data 1*). Time-lapse recordings revealed that RAB-11.1 displayed a nearly identical movement pattern to that of GIP-2, with both proteins moving together in the growth cone (*Figure 3F and G*). A frame-to-frame analysis of RAB-11.1 and GIP-2 distribution showed high correlation in the growth cone (*Figure 3—figure supplement 1F*), but no correlation in the cell body (*Figure 3—figure supplement 1G*; *Costes et al., 2004*; *Manders et al., 1993*). Co-localization and co-trafficking of RAB11.1 and GIP-2 suggest that Rab11-endosomes may be at the core of the dgMTOC.

Interestingly, Rab11-associated endosomes and even Rab11 itself have been found to influence the MT organization of mitotic spindles (*Hehnly and Doxsey, 2014*). We therefore interfered with RAB-11.1 function using two different approaches and assessed changes to γ-TuRC localization and MT polarity. Expression of a dominant negative RAB-11.1(S25N) (*Ren et al., 1998*) caused a loss of GIP-2 puncta from the dendritic growth cone in ~35% of animals (*Figure 3H and J*, n = 28, *Figure 3—source data 1*) and a reversed or mixed MT polarity in ~10% of animals (*Figure 3—figure supplement 1H*, n = 28). We speculate that some undetectable GIP-2 may remain in some growth cones in the 35% of RAB-11.1(S25N) overexpressing worms explaining a less frequent disruption of MT polarity than GIP-2 localization. As an alternative approach to test the function of RAB-11.1 in dgMTOC organization or localization, we generated Lox sites around the endogenous *rab-11* gene and expressed *Cre* specifically in PVD to knock out *rab-11* cell-specifically. 20% of *Cre* expressing worms displayed a loss of GIP-2 puncta in the dendritic growth cone with mislocalization of GIP-2 as either multiple dim puncta dispersed within the cell body (12%, *Figure 3J*,i1 and i2 in *Figure 3I*, n = 41) or along the dendrite shaft (7%, i3 in *Figure 3I*), suggesting transportation or clustering defects.

The somewhat low penetrance effects on GIP-2 localization we found with both approaches to interfering with RAB-11.1 function may be due to multiple factors. Expression in the PVD lineage may not be sufficiently early and a small variable amount of functional endogenous RAB-11.1 may

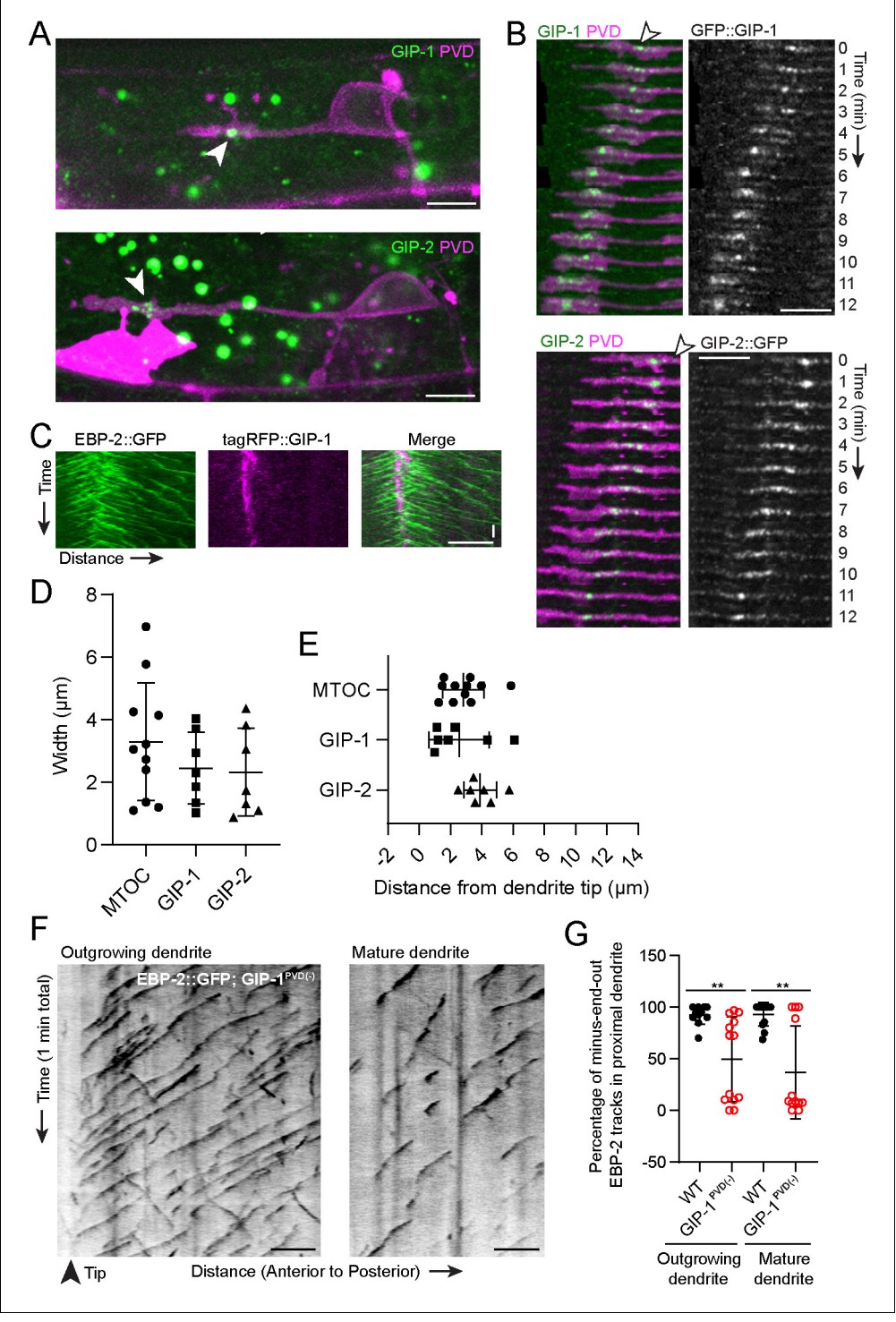

**Figure 2.** γ-TuRC localizes to the growth cone region and is required for minus-end-out MT polarity in the dendrite. (**A**) Endogenously tagged GFP::GIP-1 (top) and GIP-2::GFP (bottom) localization in an outgrowing PVD dendrite. White arrowhead, GIP cluster Unrelated structures outside of PVD include: gut granules (large green spots), HSN cell body (bright magenta region) (**B**) GIP-1 (top) and GIP-2 (bottom) localization at different time points during live imaging. White arrowhead, GIP cluster. (**C**) Kymograph of EBP-2::GFP and tagRFP::GIP-1 in the growth cone region, horizontal scale bar, 10 s (**D–E**) Quantification of the width measured as the distance between the most distal and proximal regions containing the indicated activity or protein (**D**) and shortest distance from the GIP or MTOC region to the dendrite tip (**E**). MTOC region was identified using EBP-2-GFP kymographs as in

*Figure 2 continued on next page*

*Figure 2 continued*

*Figure 1A*. Error bars represent the standard deviation (SD). (F) Kymograph of EBP-2::GFP following PVD-specific depletion of GIP-1 in an outgrowing (left) and mature (right) dendrite. (G) Quantification of MT polarity in the proximal dendrite following PVD-specific GIP-1 depletion (outgrowing dendrite, wt: n = 11 individual animals, GIP-1[PVD(-)]: n = 13 individual animals; mature dendrite, wt :n = 13 individual animals, GIP-1[PVD(-)]: n = 12 individual animals). Scale bar, 5 μm. **p<0.01, p=0.0028 for left panel and p=0.0011 for right panel, unpaired Student's *t*-test with Welch's correction, error bars represent SD. All images are lateral views oriented with anterior to the left and ventral down.

The online version of this article includes the following source data and figure supplement(s) for figure 2:

**Source data 1.** Quantification data for *Figure 2* and *Figure 2—figure supplement 1*.
**Figure supplement 1.** γ-TuRC only localizes to outgrowing anterior dendrites and is conserved among different neurons in *C. elegans*.

---

remain. Additionally, Rab11 itself may not be required for the biogenesis of these endosomes but rather play a role in their function, such as transportation (*Hehnly and Doxsey, 2014*), and may act in combination with other partially redundant proteins to correctly localize the dgMTOC. Taken together, these results suggest that γ-TuRC localization to RAB-11.1-positive endosomes in the growth cone underlies the dgMTOC, which generates short plus-end-out MTs in the distal growth cone and dynamic minus-end-out MTs that populate the growing dendrite.

## UNC-116/Kinesin-1 transports the dgMTOC through stereotyped movements

Given the specific positioning of the dgMTOC, we predicted that the subcellular localization of the dgMTOC to the dendritic growth cone is instructive to populate the proximal dendrite with minus-end-out MTs. We tested this hypothesis by perturbing the localization of the dgMTOC. We previously showed that the mature PVD anterior dendrite loses minus-end-out MTs and gains plus-end-out MTs in *unc-116/*kinesin-1 mutants (*Taylor et al., 2015*). Consistent with the MT polarity defect in mature PVD dendrites, *unc-116(e2310)* mutants showed a large decrease in the percentage of minus-end-out MTs in the anterior dendrite shaft during outgrowth (wt: 93.8 ± 2.7% (SEM, n = 11); *unc-116(e2310)*: 21.4 ± 10.4% (SEM, n = 12); *Figure 4A and B*, *Figure 4—source data 1*). This dendritic MT polarity defect in *unc-116* mutants was also evident for GFP::TBA-1 dynamics (*Figure 4—figure supplement 1B*). Additionally, the reduced minus-end-out frequency in *unc-116* mutants was partially rescued by expressing a wt *unc-116(+)* transgene during early neurite outgrowth (using the *unc-86* promoter), but not rescued by expressing *unc-116(+)* after anterior dendrite outgrowth (using a *ser-2* promoter), suggesting that UNC-116/Kinesin-1 is required specifically during early neurite outgrowth to establish dendritic minus-end-out MTs (*Figure 4—figure supplement 1A*, *Figure 4—source data 1*).

Given that UNC-116 functions during an early developmental stage and that *unc-116* mutants display MT polarity defects during dendrite outgrowth, we considered that UNC-116 might be required for dgMTOC localization or function. To test this hypothesis, we examined EBP-2 dynamics in the outgrowing dendritic growth cone where the dgMTOC normally resides. In striking contrast to wt, *unc-116* mutants lacked a convergence of plus-end-out and minus-end-out MTs in the outgrowing dendritic growth cone. Instead, the vast majority of MTs originated from the cell body and were plus-end-out throughout the anterior dendrite (*Figure 4A*, *Figure 4—figure supplement 1C and D* and *Video 4*), raising the possibility that the dgMTOC is inactive or mislocalized in *unc-116* mutants. By tracing the origin of EBP-2 comets over time in *unc-116* mutants, we frequently observed that many EBP-2 comets emanated from a single region within the cell body indicating a putative MTOC (*Figure 4C* and *Video 5*) – a stark contrast to the lack of a point-of-origin for EBP-2 comets in GIP-1[PVD(-)] animals (*Video 3*). These results suggested that the dgMTOC remains in *unc-116* mutants but is mislocalized

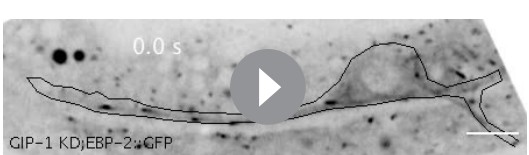

**Video 3.** EBP-2::GFP comets in the outgrowing PVD dendrite of a GIP-1[PVD(-)] animal shows plus-end-out MTs.
https://elifesciences.org/articles/56547#video3

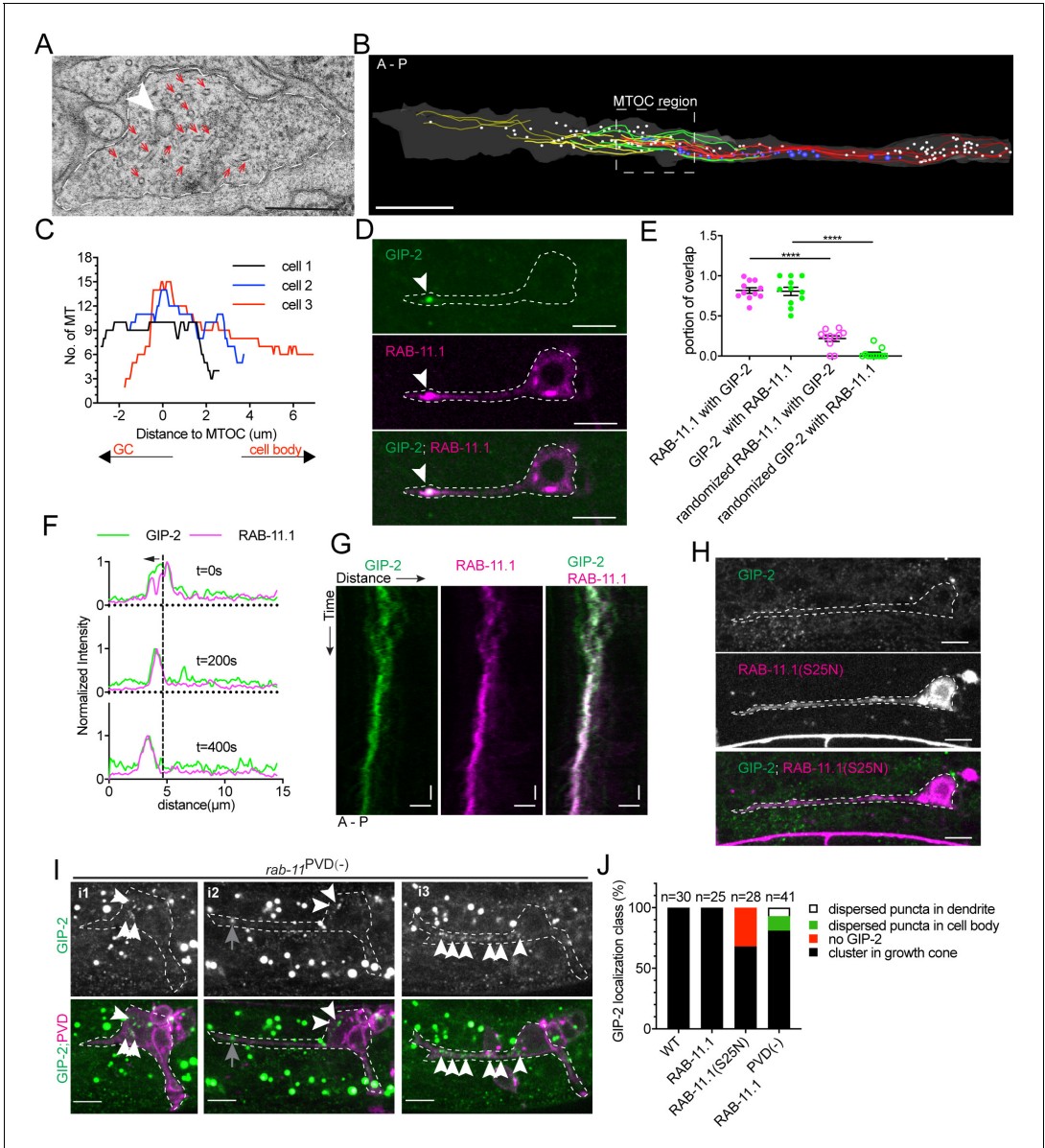

**Figure 3.** γ-TuRC localizes to RAB-11.1 endosomes. (**A**) Electron micrograph of the dendritic growth cone. Red arrows, MTs; white arrowhead, clear-core vesicle; scale bar, 0.2 µm. (**B**) Reconstruction of serial electron microscopy sections through the anterior dendrite. Yellow lines, MTs in the distal dendrite; green lines, MTs crossing the predicted MTOC region; red lines, MTs in the proximal region; white circles, clear-core vesicles; blue circles, dense-core vesicles; scale bar, 1 µm. (**C**) MT number distribution determined by serial electron microscopy in the anterior dendrite growth cone region in three PVD neurons during development. (**D**) GIP-2 and RAB-11.1 colocalization in the dendritic growth cone region. Scale bar, 5 µm. (**E**) Quantification of GIP-2 and RAB-11.1 fluorescence overlap in the growth cone region (n = 11 individual animals). ****p<0.0001, Brown-Forsythe and Welch ANOVA test, error bars represent SEM. (**F**) Normalized intensity of GIP-2 and RAB-11.1 from the dendritic tip along the dendrite shaft to the cell body at different time points. Black arrow, direction of GIP-2 and RAB-11.1 movement. Vertical dashed line, the center of GIP-2 and RAB-11 endosomes at t=0s. (**G**) Kymograph of GIP-2::GFP and mCherry::RAB-11.1 in the growth cone region. Horizontal scale bar, 2 µm; vertical scale bar, 10 s. (**H**) GIP-2 localization in worms overexpressing RAB-11.1(S25N) dominant negative mutant. (**I**) GIP-2 localization in *rab-11*^PVD(-) worms (P*unc-86::Cre; rab-11.1* (*wy1444*[*lox*])): multiple dim GIP-2 puncta in cell body (i1 and i2), dispersed dim GIP-2 puncta along the dendrite shaft (i3). Dashed white lines: PVD outline; white arrows, GIP-2 puncta; gray arrows, unrelated signal from gut granules. (**J**) Quantification of GIP-2::GFP class of localization in wt worms, worms overexpressing a RAB-11, RAB-11(S25N) dominant negative mutant or *rab-11*^PVD(-) worms (P*unc-86::Cre; rab-11.1*(*wy1444*[*lox*])). Scale bar, 5 µm. A, anterior; P, posterior. Images in D and H were taken in the *glo-1*(*zu391*) mutant background to reduce the gut granule signal.

The online version of this article includes the following source data and figure supplement(s) for figure 3:

**Source data 1.** Quantification data for *Figure 3* and *Figure 3—figure supplement 1*.

**Figure supplement 1.** Synaptic vesicles and Golgi stacks and outposts do not localize to the growth cone region.

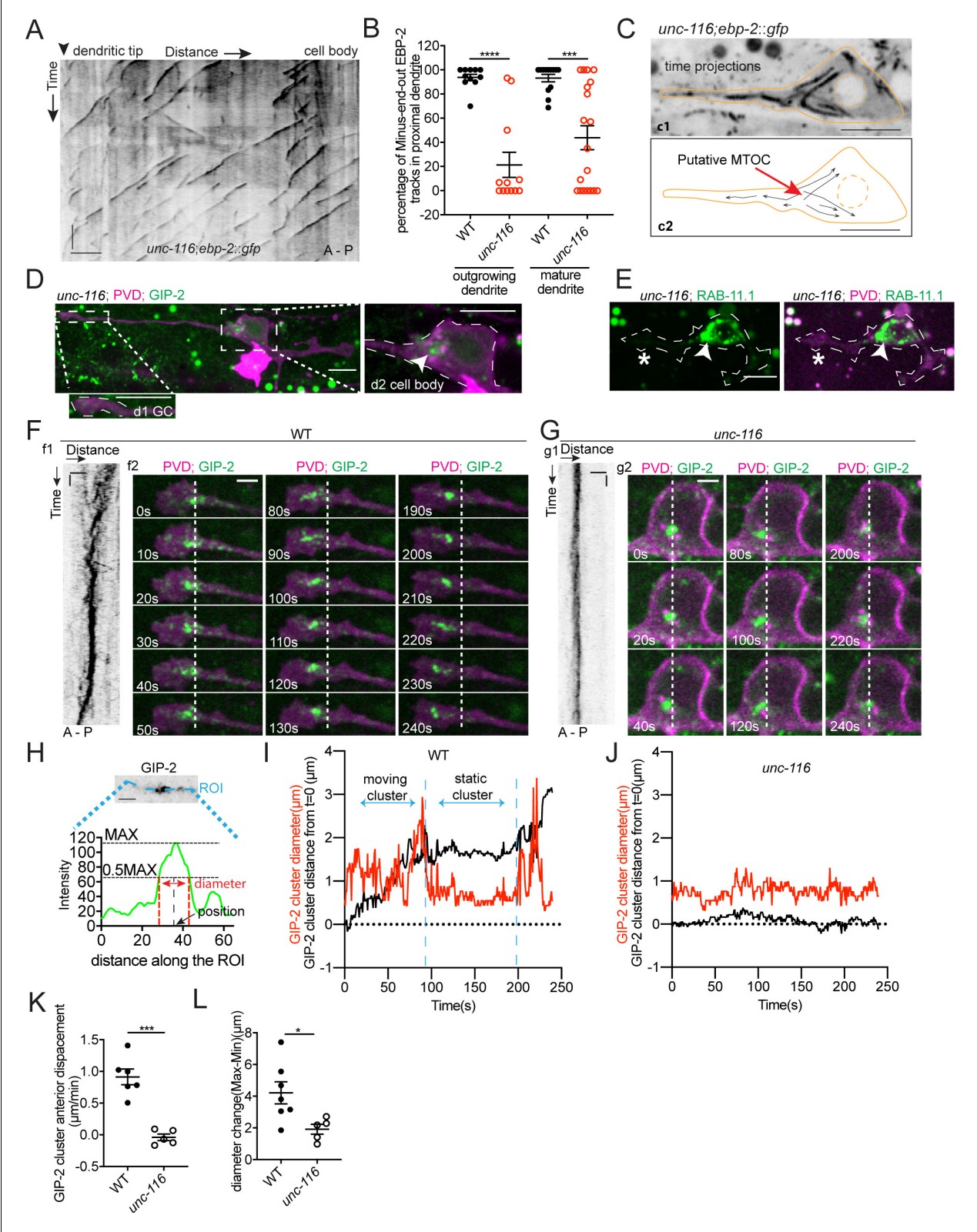

**Figure 4.** UNC-116/Kinesin-1 transports the dgMTOC anteriorly through stereotyped movements. (**A**) Kymograph of EBP-2::GFP in the outgrowing dendrite of an *unc-116* mutant. (**B**) Quantification of MT polarity in wt (n = 11 for outgrowing dendrite, n = 13 for mature dendrite) and *unc-116* mutants (n = 12 for outgrowing dendrite, n = 19 for mature dendrite). ****p<0.0001, ***p=0.0001, unpaired Student's *t*-test with Welch's correction, error bars represent SEM. (**C**) Top (**c1**): Time projection of EBP-2::GFP dynamics in an *unc-116* mutant. Bottom (**c2**): Schematic of the EBP-2::GFP

*Figure 4 continued on next page*

Figure 4 continued

trajectories generated by tracing the EBP-2::GFP frame by frame. Orange lines: the outline of the outgrowing PVD; dashed orange circles: nucleus; black arrows: EBP-2::GFP trajectories. (D) GIP-2::GFP localization in PVD in an *unc-116* mutant. Growth cone (GC) region (d1); cell body region (d2); Arrowhead: GIP-2::GFP cluster in cell body; white dotted line in d2: PVD outline. (E) GFP::RAB-11.1 localization in PVD in an *unc-116* mutant. Arrowhead: GFP::RAB-11.1 cluster in cell body; asterisk: no GFP::RAB-11.1 enrichment at PVD growth cone region; white dotted lines: PVD outline. (F–G) GIP-2::GFP dynamics in the growth cone region. Wild type (E); *unc-116* mutant (F); kymograph of GIP-2::GFP (f1 and g1); GIP-2::GFP cluster in different time points (f2 and g2); dashed lines: the position of GIP-2 cluster at t=0s.(H) Schematic of GIP-2 diameter and position for quantification. Green line: GIP-2::GFP intensity along the blue ROI; red dashed lines: the edges of the GIP-2::GFP cluster used for diameter quantification; black arrow points to the black dashed line centered between the two red dashed lines, the position of GIP-2::GFP cluster. (I–J) GIP-2::GFP cluster diameter (red line) and distance from t = 0 (black line) during GIP-2 cluster movement. Wild type (I); *unc-116* mutant (J). (K) Quantification of GIP-2::GFP cluster anterior displacement in wt (n = 6) and *unc-116* mutants (n = 5). p=0.0003, unpaired Student's *t*-test with Welch's correction, error bars represent SEM. (L) Quantification of GIP-2 diameter change in wt (n = 7) and *unc-116* mutants (n = 5). p=0.0165, unpaired Student's *t*-test with Welch's correction, error bars represent SEM. A, anterior; P, posterior; vertical scale bar, 10 s; horizontal scale bar, 5 µm in A, C D and E, 2 µm in F-H.

The online version of this article includes the following source data and figure supplement(s) for figure 4:

**Source data 1.** Quantification data for *Figure 4*, *Figure 4—figure supplement 1*, and *Figure 4—figure supplement 2*.

**Figure supplement 1.** MT polarity and dgMTOC localization require UNC-116/kinesin-1.

**Figure supplement 2.** Distribution of GIP-2::GFP cluster distance from t = 0 (black line) and diameter (red line) during GIP-2 cluster movement in different wt (n = 4, (A), *unc-116* mutants (n = 4, (B), and *dhc-1* mutants (n = 4, (C).

---

to the cell body, a possibility we assessed by examining the localization of GIP-2 and RAB-11.1. Unlike in wt, GIP-2 was largely absent from the outgrowing dendritic growth cone and instead formed a cluster within the cell body in 64% of *unc-116* mutants (n = 45, *Figure 4D* and *Figure 4—figure supplement 1E*, *Figure 4—source data 1*). RAB-11.1 also formed a cluster within the cell body rather than in the growth cone region in 71% of *unc-116* mutants (n = 28, *Figure 4E* and *Figure 4—figure supplement 1F*, *Figure 4—source data 1*), providing further support that Rab11 endosomes are a component of the dgMTOC. Occasionally, we found animals with a GIP-2 cluster in the posterior dendrite (*Figure 4—figure supplement 1G*), consistent with a rarely observed posterior dendrite-localized MTOC in *unc-116* mutants (*Figure 4—figure supplement 1H*). Together, these results indicate that the kinesin-1 motor is required for localizing the dgMTOC to the outgrowing dendritic growth cone. Furthermore, while depletion of GIP-1 in PVD showed that the dgMTOC is required to establish dendritic minus-end-out MTs, this *unc-116* mutant phenotype suggests that the location of the dgMTOC instructs MT polarity.

Next, we explored the mechanisms by which kinesin-1 localizes the dgMTOC. Time-lapse analyses of GIP-2 cluster movement showed that GIP-2 fluorescence undergoes processive movements toward the distal tip with several features. First, GIP-2 movements are largely unidirectional toward the distal dendrite tip, which is evident in kymograph analysis (f1 in *Figure 4F*), GIP-2 cluster localization at different time points (left and right panels of f2 in *Figure 4F*) and quantification of the anterior displacement of the GIP-2 cluster (*Figure 4K*). Second, GIP-2 localization alternates between a tight cluster and a broader cluster with more dispersed puncta (middle and right panels of f2 in *Figure 4F*). Third, the movements are interspersed by pauses (middle panel of f2 in *Figure 4F*), which is also evident by the alternating slopes and plateaus in GIP-2 moving distance analyses (black trace in *Figure 4I* and *Figure 4—figure supplement 2A*, *Figure 4—source data 1*). Most GIP-2 exists in a bright cluster during pauses, while the fluorescence is dispersed into multiple smaller puncta during the mobile phase (f2 in *Figure 4F* and *Video 6*). This phenomenon can be measured by the periodical changes of the diameter of the GIP-2 cluster during the movement cycle (*Figure 4H*, red trace in *Figure 4I* and *Figure 4— figure supplement 2A*, *Figure 4—source data 1*). These results indicate that GIP-2 moves in a saltatory manner, with cycles of aggregation and dispersal, toward the distal dendritic tip. We hypothesized that kinesin-1 moves the GIP-2-positive endosomes toward the distal dendritic tip along the plus-end-out MTs in the growth cone. Consistent with this idea, *unc-116*/kinesin-1 mutants showed largely reduced movement or

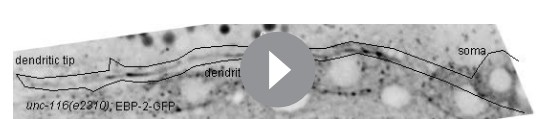

**Video 4.** EBP-2::GFP comets in the outgrowing PVD dendrite of an *unc-116(e2310)* mutant shows predominantly plus-end-out MTs.
https://elifesciences.org/articles/56547#video4

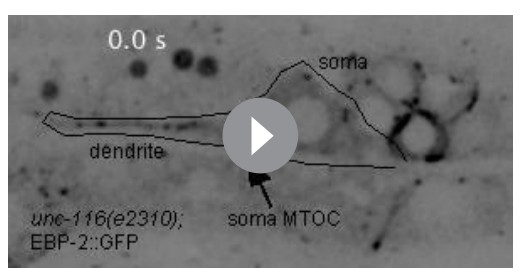

**Video 5.** EBP-2::GFP comets in the cell body of PVD in an *unc-116* mutant during dendrite outgrowth suggests an MTOC is mislocalized to the cell body.
https://elifesciences.org/articles/56547#video5

**Video 6.** GIP-2::GFP shows a stereotyped movement in the outgrowing PVD dendrite.
https://elifesciences.org/articles/56547#video6

dispersal of GIP-2 clusters (*Figure 4G and J–L*, *Figure 4—figure supplement 2B* and *Video 7*, *Figure 4—source data 1*). These analyses suggest that UNC-116/kinesin-1 moves GIP-2 containing endosomes in the outgrowing dendrite.

## UNC-116/Kinesin-1 transports the dgMTOC on transiently stabilized plus-end-out MTs

With MTs emanating from the dgMTOC in both directions, the unidirectional movements of GIP-2 suggest that kinesin-1 prefers transport on the plus-end-out MTs stretching toward the tip of the dendritic growth cone rather than the minus-end-out MTs growing toward the cell body. To investigate differences between the populations of MTs originating from the dgMTOC, we examined TBA-1 dynamics in the outgrowing dendrite (*Figure 5A*). Interestingly, we noticed that unlike minus-end-out MTs which transitioned rapidly between polymerization and depolymerization (*Figures 1C and D* and a3 in *Figure 5A*, triangle-shaped peaks in kymographs, *Figure 5—source data 1*), the individual plus-end-out MTs displayed frequent pauses between polymerization and depolymerization that lasted 10.34 ± 0.85 s (SEM, n = 34) (a2 in *Figure 5A*, plateaus, *Figure 5B and C*, *Figure 5—source data 1*), indicating that the distal plus-end-out MTs are transiently stabilized, likely through interaction with the growth cone. We note that the distal plus-end-out MTs originating from the cell body in *unc-116* mutants also paused in the tip of the dendrite and showed similar pause frequency and time as wt animals (*Figure 5B and C*, *Figure 5—source data 1*), indicating that the pausing of plus-end-out MTs in the growth cone is independent of UNC-116 function. These data reveal an asymmetry in the behavior of dgMTOC-generated MTs, with plus-end-out MTs pausing more than minus-end-out MTs.

Kinesin-1 has been shown to prefer transporting cargoes on stable over dynamic MTs (*Cai et al., 2009*; *Konishi and Setou, 2009*; *Tas et al., 2017*). To assess whether a preference of kinesin-1 for stable MTs could explain the biased dgMTOC movements, we simultaneously recorded TBA-1 dynamics and RAB-11.1 movements and assessed whether endosome movement corresponded with the presence of paused rather than dynamic MTs. No obvious distal directed RAB-11.1 movements were observed when only dynamic plus-end-out MTs were present (d1 in *Figure 5D*). In contrast, endosome movements toward the distal dendritic tip were observed in the presence of stable plus-end-out MTs (d2 in *Figure 5D*, orange lines). To correlate the RAB-11.1 movements with MT dynamics, we quantified the displacement of RAB-11.1 over time and determined and labeled time periods which have paused plus-end-out MT(s) (red, *Figure 5E* and *Figure 5—figure supplement 1A*) and periods which only have dynamic plus-end-out MTs (black, *Figure 5E* and *Figure 5—figure supplement 1A*). We found that endosome movements preferentially occurred during periods with paused MTs (red portion of solid line in *Figure 5E* and red portion of *Figure 5—figure supplement 1A*) and there was no obvious RAB-11.1 movement when the plus-end-out MTs were dynamic (dashed line in *Figure 5E* and black portion in *Figure 5—figure supplement 1A*, *Figure 5—source data 1*). By sampling of a population of worms, we found that RAB-11.1 endosome moving distance showed a positive correlation with the total MT pause time overall (*Figure 5F*, *Figure 5—source data 1*). These results suggest that distally-directed endosomes move on paused or stable

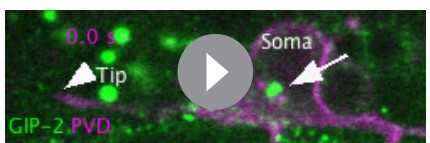

**Video 7.** GIP-2::GFP does not move in an *unc-116* (*e2310*) mutant.
https://elifesciences.org/articles/56547#video7

plus-end-out MTs in a kinesin-1-dependent manner.

Taken together, these findings are consistent with a model in which kinesin-1 transports γ-TuRC-positive endosomes toward the dendritic tip as the growth cone advances. The γ-TuRC at the dgMTOC builds plus-end-out MTs toward the growth cone, which serve as tracks for the next bouts of kinesin-1-mediated endosome movements. Individual plus-end-out MTs are stable for about 10 s followed by depolymerization events (a2 in *Figure 5A*). The transient nature of these 'stabilized' plus-end-out MTs is needed to ensure that the majority of MTs are still minus-end-out once the MTOC has passed this area.

## DHC-1/Dynein clusters γ-TuRC to form a single dgMTOC during outgrowth

While kinesin-1-mediated motility explains the processive movements of the dgMTOC towards the dendritic tip, it is not clear how the multiple γ-TuRC puncta aggregate to re-form a single cluster after their dispersal during movement, which is likely critical to maintain a singular dgMTOC. Cytoplasmic dynein has been shown to concentrate organelles such as Golgi or Rab11 endosomes near MTOCs in non-neuronal cells (*Corthésy-Theulaz et al., 1992*; *Horgan et al., 2010*). We therefore tested whether dynein is required for dgMTOC clustering by examining *dhc-1*/dynein heavy chain mutants. *dhc-1(or195)*, a temperature-sensitive dynein mutant, displayed several types of defects in dgMTOC localization when animals were cultured at the nonpermissive temperature. In ~30% of *dhc-1* mutants, GIP-2 clusters were completely absent from the PVD anterior dendrite and the rest of the cell (compare wt localization in *Figure 6A* to b1 in *Figure 6B*, *Figure 6C*, *Figure 6—source data 1*). In another ~40% of mutant animals, one or several GIP-2 cluster(s) could be found in the cell body and occasionally in the posterior dendrite (b2 in *Figure 6B*, *Figure 6C*). In the remaining ~30% of mutant animals, a GIP-2 cluster still localized to the anterior dendrite but showed a variable distance to the dendritic tip which was longer than in wt animals (b3 in *Figure 6B*, *Figure 6C and D*, *Figure 6—source data 1*). Time-lapse imaging analysis of GIP-2 movements in the subset of *dhc-1* mutant worms which have an anterior dendrite-localized GIP-2 cluster showed that net GIP-2 movement toward the dendritic tip was reduced compared to that of wt, consistent with its ectopic localization. Strikingly, further analyses of *dhc-1* mutants showed that GIP-2 puncta exhibited repeated dispersal and failed to form a relatively stable singular cluster (*Figure 6E–I*, *Figure 4—figure supplement 2C*, *Video 8* and *Figure 4—source data 1*). The continuous dispersal of GIP-2 clusters led to a reduction in the maximal intensity of the cluster over time which might explain the lack of GIP-2 clusters in a subset of *dhc-1* mutants (*Figure 6J and K*, *Figure 6—source data 1*). Endogenously tagged DHC-1 localized to the dgMTOC region as a cluster and like the GIP-2 clusters remained associated with the base of the advancing growth cone (*Figure 6L*). These dynamic movements closely coincided with those of RAB-11.1 foci (*Figure 6M*). This striking subcellular localization of DHC-1 further suggests that DHC-1 functions locally at the dgMTOC to move RAB-11.1 endosomes. Consistent with the defect in dgMTOC localization, we found that about 50% of *dhc-1* mutants showed complete reversal of MT polarity (n1 in *Figure 6N*) while another 20% of mutants showed mixed plus-end-out and minus-end-out MTs in the anterior dendrites (n2 in *Figure 6N*) perhaps due to the dispersed GIP-2 clusters in these mutants (*Figure 6N and O*, *Figure 6—source data 1*). These results suggest that dynein uses its minus-end-directed motor activity to re-cluster γ-TuRC-positive endosomes together which had dispersed during translocation towards the distal tip of the dendritic growth cone by kinesin-1, and this clustering is required for establishing dendritic minus-end-out MT polarity.

## Discussion

Collectively, these data support a model in which MTs in the dendrite are locally generated by a γ-TuRC-based endosome-associated dgMTOC which advances with the growing dendrite by the concerted action of both kinesin-1 and dynein (*Figure 6—figure supplement 1*). This dgMTOC generates plus-end-out MTs that extend toward the growing dendrite tip and minus-end-out MTs that grow toward the cell body. The plus-end-out MTs are transiently stabilized by interaction with the growth cone and serve as the tracks on which UNC-116/kinesin-1 preferentially transports the γ-TuRC-bearing endosomes further toward the distal dendritic tip as it grows. These transport events

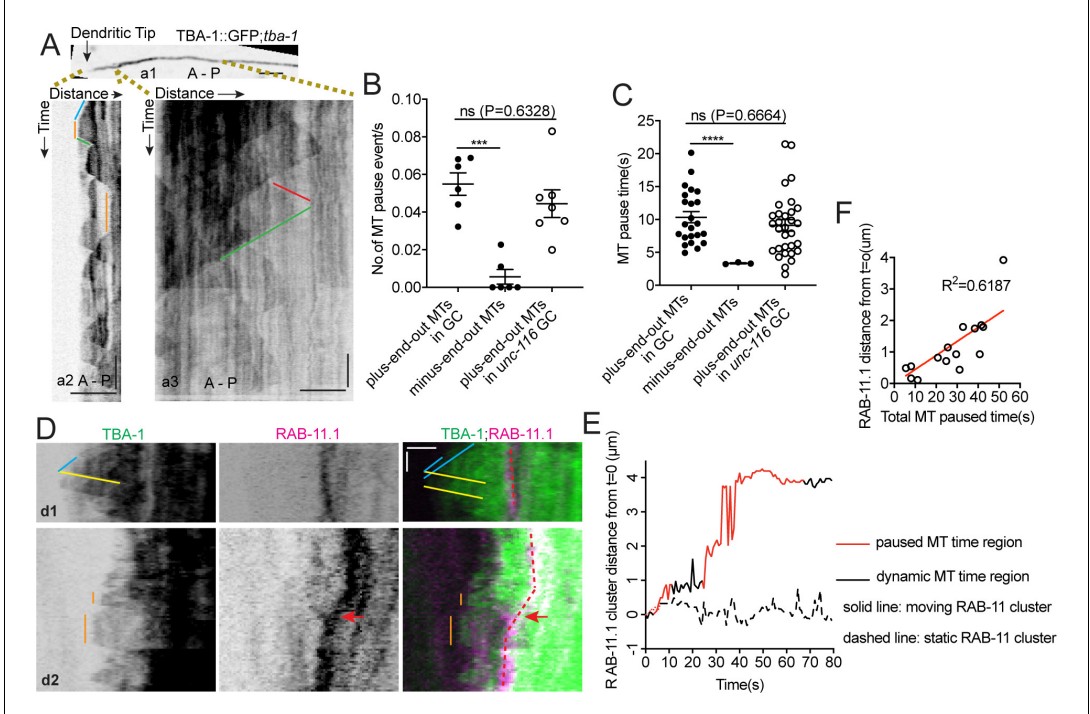

**Figure 5.** UNC-116/Kinesin-1 transports the dgMTOC on transiently stabilized plus-end-out MTs. (**A**) GFP::TBA-1 in the outgrowing dendrite (**a1**) and kymograph of GFP::TBA-1 in the indicated distal (**a2**) and proximal (**a3**) region. Blue line: growing plus-end-out MTs in distal region; red line: growing minus-end-out MTs in proximal region; green lines: retracting MTs; orange lines: pausing MTs. (**B–C**) Quantification of MT pause frequency (**B**) (n = 6 individual animals for both wt and *unc-116* mutant) and pause time (**C**) of distal plus-end-out (n = 23 for wt and n = 31 for *unc-116* mutant) and proximal minus-end-out (n = 3) individual MTs. ***p=0.0002, ****p<0.0001, Brown-Forsythe and Welch ANOVA test, error bars represent SEM. (**D**) Kymograph of GFP::TBA-1and mCherry::RAB-11.1 in the growth cone region. Blue line: growing plus-end-out MTs; yellow lines: retracting MTs; orange lines: pausing MTs. Red dashed lines indicate the position of mCherry::RAB-11.1 in the growth cone, red arrows indicate the moving mCherry::RAB-11.1 cluster. (**E**) Distribution of mCherry::RAB-11.1 cluster distance from t = 0 correlated with MT dynamics over time. (**F**) Correlation of mCherry::RAB-11.1 moving distance from t = 0 and distal plus-end-out MT pause time in different worms(n = 15). ***p<0.001, *p<0.05, unpaired Student's *t*-test, error bars represent SEM; A, anterior; P, posterior; vertical scale bar, 10 s; horizontal scale bar, 2 µm.

The online version of this article includes the following source data and figure supplement(s) for figure 5:

**Source data 1.** Quantification data for *Figure 5* and *Figure 5—figure supplement 1*.

**Figure supplement 1.** Distribution of mCherry::RAB-11.1 cluster distance from t = 0 correlated with MT dynamics over time in multiple worms.

disperse the endosomes, which are refocused by dynein-mediated minus-end-directed movements to maintain a single MTOC throughout dendrite outgrowth.

Significantly, we found that the mobile dgMTOC is required to establish the initial minus-end-out MT polarity during development that will be retained in the mature neuron for proper neuronal function. We directly observed minus-end-out MTs originating from the dgMTOC in the growth cone of the anterior dendrite, but not in the cell body or in the outgrowing axon or posterior dendrite which both lack minus-end-out MTs (*Taylor et al., 2015*), suggesting the dgMTOC lays the foundation specifically for minus-end-out MTs. Consistently, the conserved microtubule nucleating complex γ-TuRC localized to the dgMTOC and was necessary for minus-end-out MT polarity in the dendrite. Furthermore, mislocalization of the dgMTOC in both kinesin-1 and dynein mutants resulted in a lack of minus-end-out MTs in the dendrite. In fact, the presence of the dgMTOC appears to be instructive for minus-end-out MTs since a small portion of kinesin-1 mutants mislocalized the dgMTOC to the posterior dendrite and had ectopic minus-end-out MTs in the posterior dendrite.

The minus-end-out orientation of dendritic MTs initially established by the dgMTOC is likely maintained by additional mechanisms. Both EM reconstruction and TBA-1/α-tubulin dynamics in the primary dendrite show that minus-end-out MTs form a short-staggered array, which suggests that MTs do not always remain associated with the dgMTOC during outgrowth. Instead, MTs are released from the dgMTOC, raising the question of how these MTs are subsequently stabilized. Recently,

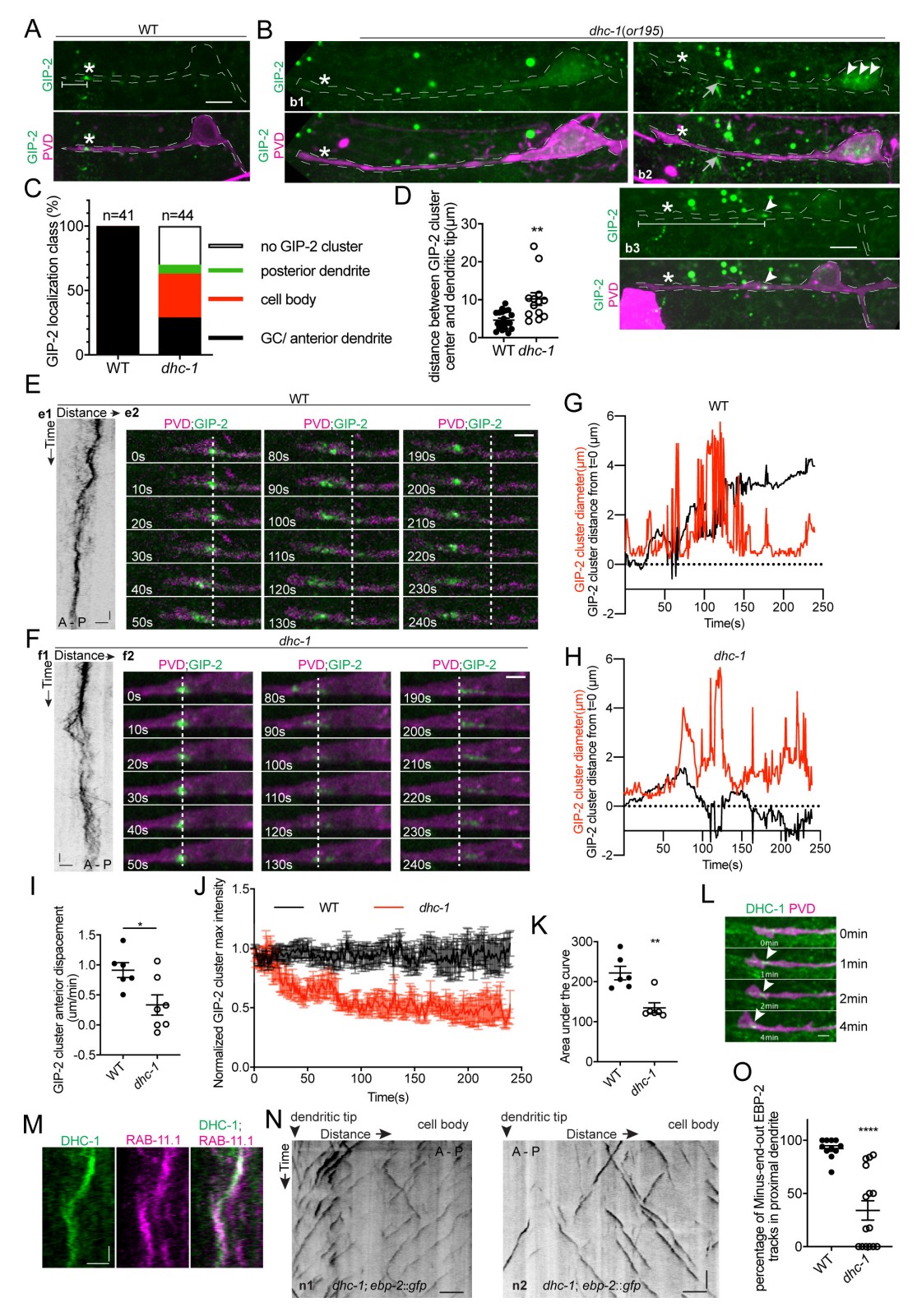

**Figure 6.** DHC-1/dynein clusters GIP-2 to form a single dgMTOC during outgrowth. (**A–B**) Endogenous GIP-2::GFP localization in wild type (**A**) and *dhc-1* mutants (**B**). White asterisks, GIP-2::GFP clusters in wt animal and growth cone region in *dhc-1* mutant; White arrowheads, GIP-2::GFP clusters in *dhc-1* mutant; gray arrows in b2, unrelated signal from gut granules; white brackets, distance between GIP-2 cluster and dendritic tip. (**C**) Quantification of GIP-2::GFP class of localization in *dhc-1* mutants. (**D**) Quantification of distance between GIP-2::GFP cluster and dendritic tip in wt (n = 20) and *dhc-1*

*Figure 6 continued on next page*

*Figure 6 continued*

mutants (n = 13). \*\*p=0.0059, unpaired Student's *t*-test with Welch's correction, error bars represent SEM. (**E–F**) GIP-2::GFP dynamics in wt (**E**) and *dhc-1* mutants (**F**). Kymograph of GIP-2::GFP (**e1** and **f1**); GIP-2::GFP cluster at different time points (**e2** and **f2**); dashed lines, the position of GIP-2 cluster at the beginning.(**G–H**) Distribution of GIP-2::GFP cluster diameter (red line) and distance from t = 0 (black line) during GIP-2::GFP cluster movement in wt (**G**) and *dhc-1* mutant (**H**). (**I**) Quantification of GIP-2::GFP cluster anterior displacement in wt (n = 6) and *dhc-1* mutants (n = 7). \*p=0.0185, unpaired Student's *t*-test with Welch's correction, error bars represent SEM. (**J**) Quantification of GIP-2::GFP cluster maximum intensity in wt (n = 7 , black) and *dhc-1* mutants (n = 7, red). (**K**) Quantification of the area under the lines in J). \*\*p=0.0025, unpaired Student's *t*-test with Welch's correction, error bars represent SEM. (**L**) Endogenous DHC-1::GFP localization in an outgrowing dendrite at different time points. White arrowheads, DHC-1::GFP in the growth cone. (**M**) Kymograph of DHC-1::GFP and mCherry::RAB-11.1 in the growth cone region. (**N**) Kymograph of EBP-2::GFP in a *dhc-1* outgrowing anterior dendrite. n1, worm with a reversed MT polarity; n2, worm with a mixed MT polarity. (**O**) Quantification of MT polarity in wt (n = 11) or *dhc-1* mutants (n = 15). \*\*\*\*p<0.0001, unpaired Student's *t*-test with Welch's correction, error bars represent SEM. Horizontal scale bar, 10 s. Vertical scale bar, 5 μm for A, B and N; others, 2 μm. Both e2 and f2 images were corrected using simple ratio method with ImageJ Bleach correction plugin. The online version of this article includes the following source data and figure supplement(s) for figure 6:

**Source data 1.** Quantification data for *Figure 6*.
**Figure supplement 1.** Proposed model for the establishment of minus-end-out MT polarity by a dendritic growth cone tracking MTOC.

cortical anchoring of MTs through an Ankyrin-CRMP complex has been showed to immobilize MTs to regulate MT stability and polarity in mature dendrites (*He et al., 2020*). Future studies will be needed to understand how this Ankyrin-CRMP-mediated anchoring mechanism functions together with the dgMTOC to create stable MTs in dendrites.

Minus-end-out MT polarity of the entire dendritic MT array is maintained throughout the entire lifetime of an organism, which is likely achieved through a collaboration of multiple MT orientation-guiding mechanisms previously shown to act in later stages of development. The minus end growth mediated by Patronin in *Drosophila* neurons grows minus-end-out MTs into terminal branches, which undergo a transition from a mixed polarity MTs to predominantly minus-end-out MTs as the dendrites mature. However, minus-end-out MTs still initially develop in the primary dendrites of *patronin* mutants (*Feng et al., 2019*). In contrast, the dgMTOC is responsible for generating minus-end-out MTs in the primary dendrite of PVD as soon as the dendrite grows out before secondary branches elaborate. At a later developmental stage when the dgMTOC is absent from PVD, MTs populate a portion of the secondary branches, likely through a dgMTOC-independent mechanism (*Liu et al., 2019*).

Additionally, to maintain the minus-end-out polarity initially established by the dgMTOC, new MTs may be generated de novo through microtubule-based microtubule nucleation (*Sánchez-Huertas et al., 2016*; *Thawani et al., 2019*), Golgi-mediated MT nucleation (*Ori-McKenney et al., 2012*), nucleation at dendrite branch sites (*Nguyen et al., 2014*) mediated by a subset of early endosomes containing Wnt signaling proteins (*Weiner et al., 2020*), or from the base of cilia in ciliated neurons (*Harterink et al., 2018*). MT patterning may be maintained by steering elongating MTs along pre-existing MTs (*Weiner et al., 2016*) and mis-oriented MTs may be removed through motor-based MT transport and sliding (*Rao and Baas, 2018*). Together, these studies highlight that MT arrays are continuously sculpted during development and in mature neurons through a plethora of mechanisms to organize MTs in dendrites specific to their subcellular location and developmental stages. However, additional maintenance mechanisms alone do not compensate for the loss of the dgMTOC during development to establish minus-end-out MTs in the mature PVD dendrite.

Importantly, our data demonstrate that the minus-end-out orientation of MTs in the dendrite – which is essential for their function in polarized trafficking – is established by a remarkably mobile endosome-associated MTOC. RAB-11.1 co-localized and co-trafficked with γ-TuRC at the site of the dgMTOC in the growth cone, and RAB-11.1 mis-localized to the same location as γ-TuRC and MTOC activity in *unc-116*/kinesin-1 mutants, consistent with the model that RAB-11.1-marked endosomes provide a scaffold onto which γ-TuRC assembles to form the dgMTOC. Rab11-associated endosomes in mitotic cells can deliver spindle material to the centrosome, suggesting

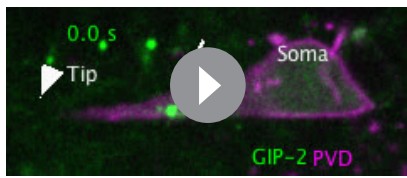

**Video 8.** GIP-2::GFP dispersion in a *dhc-1*(*or195*) mutant.
https://elifesciences.org/articles/56547#video8

that Rab11-endosomes might be a general source of MTOC material (*Hehnly and Doxsey, 2014*). Notably, interference with endogenous RAB-11.1 in PVD interfered with γ-TuRC localization to the dendritic growth cone, suggesting that RAB-11.1 itself may influence the dgMTOC. During mitosis, endosome transportation is regulated by the interaction between Rab11 and dynein (*Hehnly and Doxsey, 2014*). When we knocked out *rab-11.1* in PVD, γ-TuRC was mislocalized as dispersed puncta, similar to the dynein mutant, consistent with the idea that RAB-11.1 may interact with dynein to localize and cluster the dgMTOC. Further experiments are needed to more clearly understand the role of RAB-11.1 in dgMTOC function and explore any connection to dynein. Another outstanding question is how γ-TuRC localizes to the endosome-based dgMTOC. Several γ-TuRC tethering proteins have been found through the γ-TuRC interactome (*Paz and Lüders, 2018*; *Petry and Vale, 2015*; *Tovey and Conduit, 2018*), some of which have homologs in *C. elegans* and are candidates for future studies.

How Rab11-endosomes become active during neuronal differentiation to contribute to dgMTOC function is unclear. Dendrite outgrowth begins ~60 min after cell division, requiring a reassignment of MTOC function from the centrosomes which are active at the mitotic spindle to the dgMTOC structure during this period. Studies from epithelial cells point to a role for the centrosome in this process of MTOC reassignment (*Brodu et al., 2010*; *Sanchez and Feldman, 2017*), but it is unclear in this cell type whether the centrosome influences the genesis of this novel dgMTOC structure. Alternatively, Rab11-endosomes might act as MTOCs during mitosis and these same structures could persist following mitotic exit to form the dgMTOC. Further studies will be required to identify the origin of the dgMTOC following cell division.

Significantly, our findings together with a recent study highlight the need to more widely consider endosomes as important sources of MT nucleation in generating and maintaining the acentrosomal MT arrays found in both developing and mature dendrites. While we find that the earliest dendritic minus-end-out MTs are generated from a mobile MTOC that tracks with the dendritic growth cone and is recruited to endosomes decorated by the recycling endosome marker RAB11.1, Weiner et al. show that in *Drosophila* neurons which have already extended and branched their dendritic processes nucleate MTs at branch points from a subset of Rab5-marked early endosomes housing canonical Wnt signaling proteins (*Weiner et al., 2020*). These findings parallel in the deployment of endosomes as a platform to generate and organize MTs yet diverge in the class of the endosomes, and the location, longevity, mobility, and developmental timing of these endosomal MTOCs.

To remain associated with the advancing growth cone, the dgMTOC displays biased transport toward the distal dendrite tip mediated by kinesin-1. Distal plus-end-out MTs nucleated from the dgMTOC are transiently stabilized. A number of elegant studies have shown that kinesin-1 greatly prefers to move on stable rather than on growing MTs (*Cai et al., 2009*; *Konishi and Setou, 2009*; *Tas et al., 2017*). This preference could be mediated by differential tubulin isoforms, modifications, and/or MT-associated protein binding. It is important to note that the plus-end-out MTs at the distal dendritic growth cone are stabilized for only a short period of time (~10 s). The transiently stabilized MTs allow the dgMTOC to take its saltatory steps toward the distal dendritic tip, after which the plus-end-out MTs will depolymerize and allow the dgMTOC to achieve uniform minus-end-out MT organization behind the growth cone. How the plus-end-out MTs are transiently stabilized remains unclear but could be mediated by interaction with actin structures or membranes within the growth cone. For instance, the atypical actin motor Myo6 promotes extension of actin filament arrays in *Drosophila* sensory dendritic growth cones as they begin to branch and plus-end-out MTs polymerize specifically along these longer actin structures, supporting the idea that actin-microtubule interactions in the growth cone can influence MT dynamics and organization (*Yoong et al., 2020*).

Finally, our data and that of others support the notion that the dgMTOC is a widespread phenomenon and a generally conserved mechanism for establishing a minus-end-out MT population in primary dendrites. In a recent study looking at cytoskeletal interplay during primary dendrite branching of *Drosophila* sensory neurons, both anterograde and retrograde microtubule polymerization appear to originate from the same site in the dendritic growth cone (*Yoong et al., 2020*), as we show here, indicating the presence of a dgMTOC. Notably, these sensory neurons often have two primary dendrites both of which appear to have their own dgMTOCs, suggesting that multiple dgMTOCs can exist in different processes to establish minus-end-out MTs. Additionally, here we show that a dgMTOC is present in the outgrowing tip of different neurons in *C. elegans* as well as in *Drosophila*. In mammalian neurons, dendrites contain both plus-end-out and minus-end-out MTs.

We imagine a dgMTOC could establish the minus-end-out MT population during dendrite outgrowth in this system as well. We speculate that diverse cell types can tune other local and overall MT dynamics and maintenance programs to achieve different relative levels of plus-end-out and minus-end-out MTs to maintain a mixed polarity dendritic MT population.

# Materials and methods

## Key resources table

| Reagent type (species) or resource | Designation | Source or reference | Identifiers | Additional information |
|---|---|---|---|---|
| Strain, strain background (C. elegans) | zif-1(gk117) III; wyEx9745 [Punc86::mCherry::PLCdeltaPH] | Injected -This study | TV24185 | See C. elegans strains section in the Materials and methods |
| Strain, strain background (C. elegans) | zif-1(gk117) III; gip-1(wow5[zf::gfp::gip-1]) III; wyEx9745 | Crossed -This study | TV24455 | See C. elegans strains section in the Materials and methods |
| Strain, strain background (C. elegans) | ebp-2(wow47[ebp-2::gfp::3xflag]) II; zif-1(gk117) III; wyEx9745 | Crossed -This study | TV24458 | See C. elegans strains section in the Materials and methods |
| Strain, strain background (C. elegans) | gip-2(lt19[gip-2::gfp::loxP:: cb-unc- 119(+)::loxP]) I; wyEx9745 | Crossed -This study | TV24424 | See C. elegans strains section in the Materials and methods |
| Strain, strain background (C. elegans) | ebp-2(wow47) II; gip-1(wow25[tagRFP-t::3xMyc::gip-1]) III; wyEx9745 | Crossed -This study | TV24830 | See C. elegans strains section in the Materials and methods |
| Strain, strain background (C. elegans) | zif-1(gk117) gip-1(wow5[zf::gfp::gip-1]) III;wyEx9744[Punc-86:: mCherry::PLCdeltaPH Punc-86::zif-1] | Crossed -This study | TV24645 | See C. elegans strains section in the Materials and methods |
| Strain, strain background (C. elegans) | ebp-2(wow47 [ebp-2::gfp::3xflag]) II; zif-1(gk117) gip-1 (wow5[zf::gfp::gip-1]) III; wyEx9744[Punc-86::mCherry:: PLCdeltaPH Punc-86::zif-1] | Crossed -This study | TV24646 | See C. elegans strains section in the Materials and methods |
| Strain, strain background (C. elegans) | gip-2(lt19) I; wyIs581 [ser-2P3::myri-mCherry] | Crossed -This study | TV25492 | See C. elegans strains section in the Materials and methods |
| Strain, strain background (C. elegans) | tba-1(ok1135) II; wyIs813[Punc-86::gfp::tba-1 Punc-86:mCherry::PLCdeltaPH] | Integrated -This study | TV21720 | See C. elegans strains section in the Materials and methods |
| Strain, strain background (C. elegans) | gip-2(lt19) I; glo-1(zu391) X; wyEx10112 | Injected - this study | TV25509 | See C. elegans strains section in the Materials and methods |
| Strain, strain background (C. elegans) | gip-2(lt19) I; glo-1(zu391) X; wyEx10041[Punc-86:: mCherry::rab-11.1 cDNA Podr-1::gfp] | Injected - this study | TV25234 | See C. elegans strains section in the Materials and methods |
| Strain, strain background (C. elegans) | gip-2(lt19) I; glo-1(zu391) X; wyEx10042[Punc-86:: mCherry::RAB-11.1(S25N) Podr-1::gfp] | Injected - this study | TV25235 | See C. elegans strains section in the Materials and methods |
| Strain, strain background (C. elegans) | wyEx9876 [Punc-86::AMAN-2(1-84aa)::GFP novo2 Punc-86::mCherry:: PLCdeltaPH Podr-1::gfp] | Injected - this study | TV24622 | See C. elegans strains section in the Materials and methods |
| Strain, strain background (C. elegans) | wyEx10015[Punc-86::gfp::rab-6.2 Punc-86::mCherry ::rer-1 Podr-1::gfp] | Injected - this study | TV25150 | See C. elegans strains section in the Materials and methods |
| Strain, strain background (C. elegans) | glo-1(zu391) X; wyEx10092[Punc-86::gfp:: rab-11.1 cDNA Punc-86:: mCherry::rab-6.2 Podr-1::gfp] | Injected - this study | TV25463 | See C. elegans strains section in the Materials and methods |

*Continued on next page*

*Continued*

| Reagent type (species) or resource | Designation | Source or reference | Identifiers | Additional information |
|---|---|---|---|---|
| Strain, strain background (*C. elegans*) | *ebp-2(wow47)* II; *unc-116(e2310)* III | Crossed -This study | TV23841 | See *C. elegans* strains section in the Materials and methods |
| Strain, strain background (*C. elegans*) | *ebp-2(wow47)* II; *unc-116(e2310)* III; *wyEx9745* | Crossed -This study | TV24433 | See *C. elegans* strains section in the Materials and methods |
| Strain, strain background (*C. elegans*) | *gip-2(lt19)* I; *unc-116(e2310)* III; *wyEx9745* | Crossed -This study | TV24434 | See *C. elegans* strains section in the Materials and methods |
| Strain, strain background (*C. elegans*) | *unc-116(e2310)*; *ebp-2(wow47)*; *wyEx10049*[P*unc-86*::*unc-116* (1 ng/ul) P*unc-86*::*mCherry*:: PLC*delta*PH P*odr-1*::*gfp*] | Injected - this study | TV25242 | See *C. elegans* strains section in the Materials and methods |
| Strain, strain background (*C. elegans*) | *unc-116(e2310)*; *ebp-2(wow47)*; *wyEx10117*[ser-2P3::*unc-116* (20 ng/ul) ser-2P3::*mCherry* (10 ng/ul) P*odr-1*::*gfp*] | Injected - this study | TV25535 | See *C. elegans* strains section in the Materials and methods |
| Strain, strain background (*C. elegans*) | *unc-116(e2310)*; *ebp-2(wow47)*; *wyEx10104*[ser-2P3::*unc-116* (20 ng/ul) ser-2P3::*mCherry* (10 ng/ul) P*odr-1*::*gfp*] | Injected - this study | TV25476 | See *C. elegans* strains section in the Materials and methods |
| Strain, strain background (*C. elegans*) | *unc-116(e2310)*; *ebp-2(wow47)*; *wyEx10090*[ser-2P3::*unc-116*(20 ng/ul) ser-2P3:: *mCherry* (10 ng/ul) P*odr-1*::*gfp*] | Injected - this study | TV25461 | See *C. elegans* strains section in the Materials and methods |
| Strain, strain background (*C. elegans*) | *tba-1(ok1135)* I; *unc-116(e2310)* III; *wyEx8784*[P*unc-86*::*gfp*::*tba-1* P*unc-86*:*mCherry*::PLC*delta*PH] | Crossed -This study | TV21585 | See *C. elegans* strains section in the Materials and methods |
| Strain, strain background (*C. elegans*) | *tba-1(ok1135)* I; *wyIs813*; *wyEx10099*[P*unc-86*:: *mcherry*::*rab-11.1* cDNA P*odr-1*::*gfp*] | Injected - this study | TV25470 | See *C. elegans* strains section in the Materials and methods |
| Strain, strain background (*C. elegans*) | *dhc-1(or195)* *gip-2(lt19)* I; *wyEx9745* | Crossed -This study | TV25536 | See *C. elegans* strains section in the Materials and methods |
| Strain, strain background (*C. elegans*) | *dhc-1(ie28* [*dhc-1*:: degron::*gfp*]) I; *wyEx9745* | Crossed -This study | TV24721 | See *C. elegans* strains section in the Materials and methods |
| Strain, strain background (*C. elegans*) | *dhc-1(ie28)* I; *glo-1(zu391)* X; *wyEx10110* [P*unc-86*:: *mCherry*::*RAB-11.1* cDNA P*odr-1*::*gfp*] | Injected - this study | TV25507 | See *C. elegans* strains section in the Materials and methods |
| Strain, strain background (*C. elegans*) | *dhc-1(or195)* I; *ebp-2(wow47)* II; *wyEx9745* | Crossed -This study | TV25111 | See *C. elegans* strains section in the Materials and methods |
| Strain, strain background (*C. elegans*) | *wyIs22*[P*unc-86*:: *rab-3a*::*gfp* P*odr-1*::*rfp*] | *Patel et al., 2006* | TV201 | |
| Strain, strain background (*C. elegans*) | *unc-116(e2310)*;*wyEx9975* [P*unc-86*::*gfp*::*rab-11.1* cDNA P*unc-86*::*mCherry*:: PLC delta PH ] | Injected - this study | TV26124 | See *C. elegans* strains section in the Materials and methods |
| Strain, strain background (*C. elegans*) | *rab-11.1(wy1444[lox])*; *gip-2(lt19)*;*wyEx10192*[P*unc-86*::*Cre* P*lin-32*:: *mCherry* P*odr-1*::*gfp*] | cross- this study | TV26120 | See *C. elegans* strains section in the Materials and methods |
| Commercial assay or kit | In-Fusion HD Cloning System | Clontech | Cat 639645 | |
| Commercial assay or kit | T4 DNA ligase | NEB | Cat M0202L | |

*Continued on next page*

*Continued*

| Reagent type (species) or resource | Designation | Source or reference | Identifiers | Additional information |
|---|---|---|---|---|
| Recombinant DNA reagent | P*unc-86*::mCherry::PLCdeltaPH | This study | pMK41 | See Molecular biology, transgenic lines, and CRISPR section in the Materials and methods |
| Recombinant DNA reagent | P*unc-86*::ZIF-1 | This study | pMK32 | See Molecular biology, transgenic lines, and CRISPR section in the Materials and methods |
| Recombinant DNA reagent | P*mig-13*::*mCherry* | This study | pCM327 | See Molecular biology, transgenic lines, and CRISPR section in the Materials and methods |
| Recombinant DNA reagent | P*unc-86*::*mCherry*::*rab-11.1 cDNA* | This study | pLX107 | See Molecular biology, transgenic lines, and CRISPR section in the Materials and methods |
| Recombinant DNA reagent | P*unc-86*::*mCherry*::RAB-11(S25N) | This study | pLX110 | See Molecular biology, transgenic lines, and CRISPR section in the Materials and methods |
| Recombinant DNA reagent | P*unc-86*::AMAN-2 (1-84aa)::GFP novo2 | This study | pLX85 | See Molecular biology, transgenic lines, and CRISPR section in the Materials and methods |
| Recombinant DNA reagent | P*unc-86*::*gfp*::*rab-6.2 cDNA* | This study | pLX86 | See Molecular biology, transgenic lines, and CRISPR section in the Materials and methods |
| Recombinant DNA reagent | P*unc-86*::*mCherry*::*rab-6.2 cDNA* | This study | pLX90 | See Molecular biology, transgenic lines, and CRISPR section in the Materials and methods |
| Recombinant DNA reagent | P*unc-86*::*mCherry*::*rer-1cDNA* | This study | pLX104 | See Molecular biology, transgenic lines, and CRISPR section in the Materials and methods |
| Recombinant DNA reagent | P*unc-86*::*gfp*::*rab-11.1 cDNA* | This study | pLX99 | See Molecular biology, transgenic lines, and CRISPR section in the Materials and methods |
| Recombinant DNA reagent | P*unc-86*::*unc-116* | This study | pLX96 | See Molecular biology, transgenic lines, and CRISPR section in the Materials and methods |
| Recombinant DNA reagent | *ser-2*P3::*unc-116* | This study | pLX98 | See Molecular biology, transgenic lines, and CRISPR section in the Materials and methods |
| Recombinant DNA reagent | P*unc-86*::Cre | This study | pCY26 | See Molecular biology, transgenic lines, and CRISPR section in the Materials and methods |
| Sequence-based reagent | *gip-1*(*wow5*) gRNA | *Sallee et al., 2018* | | TCAGATAAATGCGTCGACAAGG |
| Sequence-based reagent | *gip-1*(*wow5*)-HA_F | *Sallee et al., 2018* | | GGAGAAAATTAACCAAAAACTT GAAATTTTATGAAAAAAAAATG GAAAAATTTCAGATAAATGCCG ACAGAATACAAAACGCGA CTTTGTGATG |

*Continued on next page*

*Continued*

| Reagent type (species) or resource | Designation | Source or reference | Identifiers | Additional information |
|---|---|---|---|---|
| Sequence-based reagent | *gip-1(wow5)*-HA_R | *Sallee et al., 2018* | | CTATCATGAAACCCGAAAGCATT TAAAAATTGCTGTACAGCTTCAA CTTCTTCGCTGCCTTGACGTCGC ATGGCTCCGCTAGCTCCTGATTT GTATAGTTCGTCCATGCCA TGTGTAATCCC |
| Sequence-based reagent | *ebp-2(wow47)* sgRNA | *Sallee et al., 2018* | | GCAGGCAAATCTGGACGATACGG |
| Sequence-based reagent | *ebp-2(wow47)* 5'HA-F | *Sallee et al., 2018* | | TTGTAAAACGACGGCCAGTCG CCGGCAGTTGCTGC TCCTGCTAGACC |
| Sequence-based reagent | *ebp-2(wow47)* 5'HA-R | *Sallee et al., 2018* | | CATCGATGCTCCTGAGGCTCCCG ATGCTCCGAAAGTCTC GGTATCGTCCAGATT |
| Sequence-based reagent | *ebp-2(wow47)* 3'HA-F | *Sallee et al., 2018* | | CGTGATTACAAGGATGACGAT GACAAGAGATAAATATTG TTGTTTCCCATTGCTT |
| Sequence-based reagent | *ebp-2(wow47)* 3'HA-R | *Sallee et al., 2018* | | GGAAACAGCTATGACCATG TTATCGATTTCTTTGCGA TTGATGATGTCGT |
| Sequence-based reagent | *gip-1(wow25)* gRNA | *Sallee et al., 2018* | | TCAGATAAATGCGTCGACAAGG |
| Sequence-based reagent | *gip-1(wow25)* 5'HA-F | *Sallee et al., 2018* | | CACGACGTTGTAAAACGACG GCCAGTCGTACTGGAAA TTTGGGCAACA |
| Sequence-based reagent | *gip-1(wow25)* 5'HA-R | *Sallee et al., 2018* | | CTTGATGAGCTCCTCTCCCT TGGAGACCATTTATCTG AAATTTTTCCATTTT |
| Sequence-based reagent | *gip-1(wow25)* 3'HA-F | *Sallee et al., 2018* | | GAGCAGAAGTTGATCAGCGA GGAAGACTTGCGTCGA CAAGGCAGCG |
| Sequence-based reagent | *gip-1(wow25)* 3'HA-R | *Sallee et al., 2018* | | TCACACAGGAAACAGCTAT GACCATGTTATCAAAATTC AAAATCCCCGTTT |
| Sequence-based reagent | *rab-11.1(wy1444)* 5'donor | This study | | gctgatgaatcatgtgaccaatgcc ctttttctttttacaatcgtcccaataa cttcgtataatgtatgctatacgaagtt atatatatacacaactttcaaacaagct ttatcattttacagctcagcagtaaag |
| Sequence-based reagent | *rab-11.1(wy1444)* 3'donor | This study | | gagtttatcgaattcttgcaagcact gcgtttgcaagtcttcaccgtttataactt cgtataatgtatgctatacgaagttatt ggtgtgtagtatttgtaactttcttt cagatttatttgtaattgcctcc |
| Sequence-based reagent | *rab-11.1(wy1444)* 5' gRNA | This study | | TTGAAAGTTGTGTATATATT GGG |
| Sequence-based reagent | *rab-11.1(wy1444)* 3' gRNA | This study | | tttgcaagtcttcaccgttttgg |
| Chemical compound, drug | Levamisol hydrochloride | Sigma-Aldrich | Cat 31742 | |
| Software, algorithm | Image J | NIH | RRID:SCR_003070 | https://imagej.net/ImageJ |
| Software, algorithm | Fiji | GitHub | RRID:SCR_002285 | https://fiji.sc/ |
| Software, algorithm | GraphPad Prism 8 | GraphPad | RRID:SCR_002798 | https://www.graphpad. com/scientific- software/prism/ |
| Software, algorithm | MetaMorph | Molecular Devices | RRID:SCR_002368 | |

### C. elegans strains

Worms were raised on NGM plates at 20°C using OP50 *Escherichia coli* as a food source. Worm strains which contain temperature sensitive mutant *dhc-1*(*or195*) were maintained in 16°C and were shift to 25°C for phenotype analysis. *C. elegans* strains used in this study are listed in the key resources table. OD2509 [*gip-2*(*lt19*[*gip-2::gfp*]::loxP::cb-unc-119(+)::loxP) I; *unc-119*(*ed3*) III] was a gift from Dr. Karen Oegema at the University of California San Diego (*Wang et al., 2017*). The generation of JLF273 *ebp-2*(*wow47*[*ebp-2::gfp*]) II; *zif-1*(*gk117*) III; *wowEx10* (used to make TV23841), JLF38 *gip-1*(*wow5*[*zf::gfp::gip-1*]) *zif-1*(*gk117*) III; *wowEx10* (used to make TV24455), and JLF155 *zif-1* (*gk117*) has been described (*Sallee et al., 2018*). Strains are available upon request.

## Electron microscopy

Worms were prepared for conventional EM by high pressure freezing/freeze-substitution. Worms in *E. coli* containing 20% BSA were frozen in 100 μm well specimen carriers (Type A) opposite a hexadecane coated flat carrier (Type B) using a BalTec HPM 01 high-pressure freezer (BalTec, Lichtenstein). Freeze-substitution in 1% OsO4, 0.1% uranyl acetate, 1% methanol in acetone, containing 3% water (*Buser and Walther, 2008*; *Walther and Ziegler, 2002*) was carried out with a Leica AFS2 unit. Following substitution, samples were rinsed in acetone, infiltrated and then polymerized in Eponate 12 resin (Ted Pella, Inc, Redding, CA). Serial 50 nm sections were cut with a Leica UCT ultramicrotome using a Diatome diamond knife, picked up on Pioloform coated slot grids and stained with uranyl acetate and Sato's lead (*Sato, 1968*). Sections were imaged with an FEI Tecnai T12 TEM at 120 kV using a Gatan 4k × 4 k camera. TrakEM2 in Fiji was used to align serial sections (*Cardona et al., 2012*; *Schindelin et al., 2012*). Modeling of serial sections was performed with IMOD (*Kremer et al., 1996*).

## Molecular biology, transgenic lines, and CRISPR

Plasmids and primers used to generate transgenic or knock in *C. elegans* strains in this study are listed in the key resources table. Detailed information on plasmid generation and primer sequences are provided in *Supplementary file 1*. Expression clones were made in the pSM vector, a derivative of pPD49.26 (A. Fire) with extra cloning sites (S. McCarroll and C.I. Bargmann, personal communication). *rab-11.1*(*wy1444*[*lox*]) was generated by co-injecting oligo donors and CAS9 RNPs (*Dokshin et al., 2018*). P*unc-86*::mCherry::PLCdeltaPH was generated using Gibson cloning. Transgenic strains (1–50 ng/μl) were generated using standard techniques and coinjected with markers P*odr-1*::GFP or P*odr-1*::RFP. Plasmids are available upon request.

### C. elegans synchronization and staging

Gravid adults were bleached in a hypochlorite solution to obtain embryos which were washed in M9 and allowed to hatch either in M9 or on unseeded NGM plates overnight to obtain a population synchronized in L1 arrest. This L1-arrested population was kept for a maximum of 5 days and transferred to OP50-seeded NGM plates at different times to achieve specifically aged synchronized L2 populations for imaging.

To image early outgrowing PVD neurites, wild type and GIP-1 knockdown L1-arrested animals were grown on OP50-seeded NGM plates at 22°C for 18–19 hr or 25°C for 16–17 hr before imaging. *unc-116*(*e2310*) L1-arrested animals were grown on OP50-seeded NGM plates at 22°C for 20–22 hr or 25°C for ~18 hr before imaging.

*dhc-1* l1-arrested worms were obtained at 25°C and then were grown on OP50 seeded NGM plates at 25°C for 16–18 hr.

Wild-type GFP::TBA-1 animals were imaged 1.5–2 hr later than other wild-type image to get a more clear MT dynamic kymograph in the growth cone region, as in the later outgrowth stage, the MT number in the growth cone region will slightly reduce.

To image mature PVD neurites, bleached adults were placed on OP50-seeded NGM plates for about 48 hr at 20°C, then the mid L4 worms were picked to image for EBP-2 dynamics and GIP-2 localization.

To image the outgrowing DA9 neuron, some 3-fold stage embryos were transferred to a new plate and then the L1 worms right after hatching were imaged.

## Slide preparation

Just prior to imaging, *C. elegans* animals were mounted on 3–5% agarose pads. L2s or L1s were picked and released into a 1 µl M9 or water droplet on an inverted coverslip while minimizing bacterial transfer. Prior to mounting on the freshly made agarose pad, the droplet was surrounded by a 1 µl droplet of 0.05 µm Polysterene Polybeads (Polysciences) and a droplet of levamisole (final concentration of approximately 3 mM) to immobilize worms. L4s were picked to M9 directly on the agarose pad with levamisole added prior to mounting. Slides were sealed with VALAP or Vaseline prior to time-lapse imaging. All imaging was performed within 40 min of mounting.

## Microscope system

Imaging of *C. elegans* was performed on an inverted Zeiss Axio Observer Z1 microscope equipped with a Yokogawa spinning disk, QuantEM:512SC Hamamatsu camera (set to 600 EM Gain), a Plan-Apochromat 100x/1.4 NA objective (Zeiss), 488 nm and 561 nm lasers, and controlled by Meta-Morph Microscopy software (Molecular Devices).

## Imaging parameters

Endogenous EBP-2::GFP dynamics in outgrowing and mature PVD dendrites were imaged using 50% 488 nm laser power, 100 ms exposure, and with 200 ms time interval between acquisitions. 200–300 frames were recorded per animal. The same imaging parameters were also used to image GFP::TBA-1 dynamics in outgrowing dendrite.

To assess protein localization within PVD during neurite outgrowth, endogenously-GFP-tagged GIP-1, GIP-2 were imaged using 70–80% 488 nm laser power recorded with 300 ms exposure. The membrane mCherry co-marker was imaged using 70–80% 561 nm laser power with 200 ms exposure. To fully sample PVD in three dimensions, z-sections were imaged at Nyquist resolution, every 0.25 µm from above to below a single PVD in each animal. Micrographs in *Figure 2A*, and *Figure 2—figure supplement 1A* are maximum projections of the z-sections that include the PVD dendrite.

For time-lapse imaging of these strains to assess dynamics of complex localization to the outgrowing dendrite tip, 488 nm laser power was reduced to 50% with 200 ms exposures to reduce phototoxicity and photobleaching. Up to 4 z-sections were imaged at 0.5 µm, and acquisitions occurred every 15 or 30 s for up to a total of 60 min. Note that *Figure 2B* displays only frames every 60 s even if animals were imaged more frequently. Only animals which displayed a steady rate of growth cone advance indicating healthy animals were used for analysis.

For time lapse imaging to look at the details of GIP-2, RAB-11.1 and DHC-1 dynamics at the growth cone region, only one z section is acquired, and images were taken every 200 ms for up to 200–300 frames, and images were taken with a 100 ms exposure time and 70% laser power for both 488 nm and 561 nm channels. To image TBA-1 together with RAB-11.1, the 488 nm laser power was reduced to 50% while other parameters were the same.

## Image analysis and quantification

Images were processed and analyzed using MetaMorph (Molecular Devices) and ImageJ to create kymographs or psuedocolored merged maximum intensity micrographs and assembled into figures using Adobe Photoshop and Illustrator. Statistical calculations and graphing were done in Prism 7 and Prism8 (GraphPad).

To display the dynamics of endogenously-GFP-tagged GIP-1 and GIP-2 localization to the PVD dendritic growth cone over time in *Figure 2B,a* 20 pixel-wide line segment was drawn over the growth cone outgrowth region and processed using the ImageJ function 'Straighten' to slightly straighten the region. The Make Montage function was then used to create a montage displaying that region every 60 s. Due to minor movements of the worm, uncropped merged time-lapse images were registered to each other using the ImageJ StackReg function before making a montage for GIP-1.

Kymographs were made in ImageJ by drawing a straight or segmented line (width of 6–10 pixels) along a process and using the KymographBuilder plugin or Stacks-Reslice function on the stacked time-lapse file.

To quantify EBP-2::GFP comet direction and frequency at interval distances across the outgrowing dendrite (line graphs in *Figure 1C*, *Figure 4—figure supplement 1C and D*), kymographs were

created from recordings of numerous animals and the number of EBP-2::GFP minus-end-out and plus-end-out tracks was manually counted at each specified distance. For *Figure 1C*, the MTOC region was first defined as the region in which the majority of both directions of comets originated, and the center of that region was designated as zero.

To quantify overall MT polarity in the outgrowing PVD dendrite, for outgrowth stage, the imaging region was the whole process while the polarity quantification was done in the proximal dendrite region which is within 50 μm from cell body and excluding the MTOC region; for the mature dendrite, the quantification is also done in the proximal dendrite.

To quantify the co-localization between GIP-2 and RAB-11.1in different worms (*Figure 3E*), a 30 × 20 pixel ROI was drawn in the growth cone region first and then was duplicated to independent images for both GIP-2 and RAB-11.1 channels, then a random GIP-2 or RAB-11.1 localization image was generated by ImageJ JACop plugin, and then the thresholded Mander's split colocalization coefficients(tM) values were measured by the Colocalization Threshold plugin which can generate the threshold automatically between GIP-2 and RAB-11.1, GIP-2 and randomized RAB-11.1, randomized RAB-11.1 and GIP-2.

To quantify the co-localization between GIP-2 and RAB-11.1 over time in the same worm, the thresholded Mander's split colocalization coefficients, Pearson's correlation coefficient (Rcoloc) and the scatterplot files were all generated by the Colocalization Threshold plugin for 100 frames in both the growth cone and cell body region.

To measure the GIP-2 cluster diameter and distance from t = 0 (*Figure 4H–J*, *Figure 6G and H*, *Figure 4—figure supplement 2A–C*), as shown in *Figure 4H,a* segmented line was draw along the GIP-2 moving track, and then the plot file for all time frames were generated in ImageJ and exported to an excel file to get the intensity value of different positions along the line and also the intensity values at different time points. Then the same segmented line was moved to a nearby region to get the background intensity value and also exported to an excel file. The final GIP-2 intensity value was the original intensity value minus the background intensity value. The pixel value of the cluster diameter at one given time point $D_t$ was the distance between the first ($P_{first}$) and last ($P_{last}$) position in which the intensity value is above the 50% maximum intensity value in the same time point. The pixel position of the cluster $P_t$ was the center of $P_{first}$ and $P_{last}$ (*Figure 3F*). And then:

$$\text{Diameter}\,(\mu m) = 0.109^{*}\,D_t$$

$$\text{Distance from}\,t = 0\,(\mu m) = -0.109^{*}(P_t - P_0)\,(0.109\,\mu m\,\text{pixel for the 100X objective calibration})$$

The RAB-11.1 distance from t = 0 (*Figure 5E and F*, *Figure 5—figure supplement 1A*) was calculated the same way as GIP-2, but the threshold was set to above 70% maximum RAB-11.1 intensity, as the mCherry::RAB-11.1 showed a higher background than GIP-2.

To measure the GIP-2 cluster maximum intensity dynamics during movement (*Figure 6J*), the background of the movie was subtracted using the ImageJ subtract background function, then the maximum intensity of both the green GIP-2 channel and red PVD morphology marker channel at different time points were measured using the ImageJ ROI multiple measure function. The green GIP-2 intensity was divided by the red PVD morphology channel intensity to correct for focus change or photo-bleaching, and the normalized GIP-2 maximum intensity at different time points we calculated by setting the start value as 1.

The area under the different maximum intensity curves were calculated using Prism.

Any movie that showed a slight movement was aligned by ImageJ StackReg plugin before analysis.

### *D. melanogaster* methods

The previously generated strain *Rluv3-Gal4, UAS-EB1::GFP* (*Yalgin et al., 2015*) was used to investigate the organization of MT minus ends in *D. melanogaster* class I da sensory neurons. Preparation of embryos was carried out as previously described (*Yalgin et al., 2015*). Briefly, stage-15 embryos were quickly de-chorionated, washed in water, and mounted with halocarbon oil with one layer of double-sided tape as a spacer. Imaging of *D. melanogaster* was performed with a FV1200 Laser Scanning Microscope (Olympus) equipped with a 473 nm laser, and a PLAPON 60XO NA1.42 (Olympus) objective. Time-lapse images were acquired at 5% laser power at 5x zoom, imaging three

z-sections spaced 0.6 μms apart, with 2.4 s between frames, for four minutes. Kymographs were prepared using ImageJ using an average projection of imaged z-sections.

## Acknowledgements

We thank Callista Yee for the providing a P*unc-86*::Cre expressing plasmid and thank members of the Shen lab for their scientific feedback and discussion.

## Additional information

### Competing interests

Kang Shen: Reviewing editor, *eLife*. The other authors declare that no competing interests exist.

### Funding

| Funder | Grant reference number | Author |
| --- | --- | --- |
| Howard Hughes Medical Institute | | Kang Shen |
| National Institutes of Health | NS082208 | Kang Shen |
| National Institutes of Health | NIH New Innovator Award: DP2GM119136-01 | Jessica L Feldman |
| National Institutes of Health | NIGMS NIH award F32GM120913-01 | Maria Danielle Sallee |
| National Institutes of Health | R01GM133950 | Jessica L Feldman |

The funders had no role in study design, data collection and interpretation, or the decision to submit the work for publication.

### Author contributions

Xing Liang, Marcela Kokes, Conceptualization, Formal analysis, Investigation, Visualization, Writing - review and editing, Methodology, Validation.; Richard D Fetter, Formal analysis, Investigation; Maria Danielle Sallee, Resources, Funding acquisition; Adrian W Moore, Investigation, Writing - review and editing; Jessica L Feldman, Conceptualization, Supervision, Funding acquisition, Writing - review and editing; Kang Shen, Conceptualization, Supervision, Funding acquisition, Writing - original draft, Project administration, Writing - review and editing

### Author ORCIDs

Xing Liang https://orcid.org/0000-0002-8298-1214
Marcela Kokes https://orcid.org/0000-0002-7218-481X
Richard D Fetter http://orcid.org/0000-0002-1558-100X
Jessica L Feldman https://orcid.org/0000-0002-5210-5045
Kang Shen https://orcid.org/0000-0003-4059-8249

### Decision letter and Author response

Decision letter https://doi.org/10.7554/eLife.56547.sa1
Author response https://doi.org/10.7554/eLife.56547.sa2

## Additional files

### Supplementary files

• Supplementary file 1. Plasmid and sequence.
• Transparent reporting form

**Data availability**

All data generated or analyzed during this study are included in the manuscript and supporting files.

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
