## [Decision Letter]

**Acceptance summary:**

This work identifies a novel, endosome-associated microtubule organizing center (MTOC) at the tip of dendrites in certain types of neurons in *C. elegans*. The work adds endosomes to a growing list of cellular structures that, apart from centrioles, can function as microtubule-nucleating and organizing structures. Through identification of this MTOC the study also provides a mechanism for how the specific orientation of microtubules in dendrites can be achieved, which is crucial for dendritic identity and function.

**Decision letter after peer review:**

Thank you for submitting your article "Growth Cone-Localized Microtubule Organizing Center Establishes Microtubule Orientation in Dendrites" for consideration by *eLife*. Your article has been reviewed by three peer reviewers, including Jens Lüders as the Reviewing Editor and Reviewer #1, and the evaluation has been overseen by Piali Sengupta as the Senior Editor.

The reviewers have discussed the reviews with one another and the Reviewing Editor has drafted this decision to help you prepare a revised submission.

Summary:

This manuscript investigates how the minus-end-out microtubule network in outgrowing dendrites is established. This is a very important question, since the difference in the polarity of microtubules in axons (uniform plus-end-out) and dendrites (mixed or minus-end-out) determines the identity of these neuronal compartments through differential sorting of cargoes. Using *C. elegans* PVD neurons as model, the authors identify a novel MTOC near the dendritic growth cone ("dgMTOC"). The authors provide evidence that the MTOC is localized at endosomes and that it generates the minus-end-out microtubules of the dendrite shaft. This MTOC also generates a few plus-end-out microtubules that are used by the MTOC to track with the growing tip. This tracking is mediated by kinesin-1, whereas dynein is required for keeping the MTOC clustered.

All reviewers agree that this work represents an important discovery combined with very well-executed mechanistic analyses. However, they consider the evidence that supports the endosomal identity of the dgMTOC not fully convincing. This could be addressed in two ways: first, by providing additional analyses, if possible, and second, by deemphasizing the endosomal identity in the text.

This issue and additional suggestions for improvement are detailed below.

Essential revisions:

1) One of the conclusions that could be better supported is the role of RAB-11 in localizing the γ-TuRC, which is currently based on correlative data only. The data presented in Figure 3H and I show that expression of dominant negative RAB-11.1 caused defective GIP-2 localization in 35% of animals, which is somewhat low. Given that this phenotype is weak, could the authors inactivate RAB-11.1 by another method and demonstrate that there are similar defects? Or provide any other evidence supporting recruitment of γ-TuRC to endosomes? Related to this, were there microtubule polarity defects in the 35% of animals with defective GIP-2 localization?

We also ask the authors, in the absence of additional evidence, to tone down text passages, figure and paragraph titles, and model suggesting direct association of gTuRC with endosomes.

2) Given the distinct phenotypes of UNC-116 and GIP-1 and as support for the depletion of GIP-1, we recommend that the number of comets in control and in these two conditions is analyzed. This might be possible on the already existing data.

---

## [Author Response]

Essential revisions:1) One of the conclusions that could be better supported is the role of RAB-11 in localizing the γ-TuRC, which is currently based on correlative data only. The data presented in Figure 3H and I show that expression of dominant negative RAB-11.1 caused defective GIP-2 localization in 35% of animals, which is somewhat low. Given that this phenotype is weak, could the authors inactivate RAB-11.1 by another method and demonstrate that there are similar defects? Or provide any other evidence supporting recruitment of γ-TuRC to endosomes? Related to this, were there microtubule polarity defects in the 35% of animals with defective GIP-2 localization?We also ask the authors, in the absence of additional evidence, to tone down text passages, figure and paragraph titles, and model suggesting direct association of gTuRC with endosomes.

We thank the reviewers for the suggestion. To further strengthen the correlation between γ-TuRC and RAB-11.1 positive endosome localization, we examined the RAB-11.1 endosome localization in *unc-116*/Kinesin1 mutants and found that RAB-11.1 endosomes accumulated in the cell body rather than in the growth cone region, which is consistent with the ectopic localization of GIP-2 in an *unc-116* mutant. We added this data to the new Figure 4E. While these new results are correlative (like the colocalization and co-movement of RAB-11.1 and GIP-2 in wt), the mislocalization of RAB-11.1, GIP-2, and MTOC activities to the same specific subcellular locus in *unc-116* mutants further supports the notion that the dgMTOC is comprised of γ-TuRC positive Rab11 endosomes.

To test the causality between endosomes and the dgMTOC, we would need to eliminate endosomes. There is not an established way to do this, so we chose instead to inactivate RAB-11.1 in an attempt to perturb endosome number, localization, and/or behavior. Our results detailed below suggest that RAB-11.1 is likely to be required for certain endosome behaviors but is not essential for the recruitment of γ-TuRC to endosomes. We would like to stress that we are not arguing that RAB-11.1 is essential to establish the dgMTOC, but rather using its inactivation as a tool to probe the relationship between endosomes and the dgMTOC.

Since *rab-11.1* is an essential gene, we relied on its tissue specific removal in our experiments in PVD. To do this, we used the *Cre/lox* system to remove the *rab-11.1* gene from cells within the PVD lineage. We inserted *lox* sites around the endogenous *rab-11.1* gene and expressed *Cre* using a cell-type specific promoter. To achieve a highly efficient knock out, we first attempted to remove *rab-11.1* from the V5 seam cell, a distant ancestor of PVD, and its descendants by expressing *Cre* using an *nhr-81* promoter. Unfortunately, this treatment was lethal likely due to the large number of cells affected and the early loss of *rab-11.1*. We next tried the *unc-86* promoter, which starts expressing just prior to the birth of PVD. Among 41 *Cre* expressing worms, 5 worms showed multiple dim GIP-2 puncta in the cell body and 3 worms showed dispersed GIP-2 puncta along the dendrite with all 8 worms lacking GIP-2 in the growth cone. We added this data to Figure 3I and J.

Both dominant negative RAB-11.1 and PVD-specific *rab-11.1* knock out showed a GIP-2 localization defect, but with a low penetrance. This low penetrance can be explained by several factors: 1) Inefficient removal of the *rab-11.1* gene due to the timing and/or robustness of *Cre* expression; 2) Perduring RAB-11.1 protein and/or mRNA – the outgrowth of the anterior dendrite takes place about one hour after PVD is born and so existing RAB-11.1 positive endosomes, protein, and/or mRNA may perdure despite the complete removal of the *rab-11.1* locus; 3) RAB-11.1 acts redundantly with other factors to localize GIP-2 and/or build endosomes. The GIP-2 localization in *rab-11.1* mutant worms is similar to that of dynein mutants, suggesting that RAB-11.1 might play a role in transporting endosomes but might not be required for recruiting γ-TuRC to endosomes. It is likely that endosome-specific tethers exist to directly recruit microtubule minus end proteins such as γ-TuRC as has been seen to underlie MTOC function at other organelles such as the centrosome and Golgi apparatus. We have discussed this point in our manuscript to clarify that the nature of any direct interaction between γ-TuRC and endosomes is unclear and mentioned potential tethering proteins in the Discussion (sixth paragraph).

Finally, we examined microtubule polarity in the dominant negative RAB-11.1 overexpression line and found that only 3 of 28 worms showed a reversed or mixed microtubule polarity and added the data to Figure 3—figure supplement 1H. The penetrance of this phenotype is even lower than the GIP-2 localization phenotype, and it is possible that a low amount of GIP-2, below our limit of detection, at the growth cone might be sufficient to establish the minus-end-out microtubule population.

2) Given the distinct phenotypes of UNC-116 and GIP-1 and as support for the depletion of GIP-1, we recommend that the number of comets in control and in these two conditions is analyzed. This might be possible on the already existing data.

We thank the reviewers for the suggestion. We quantified the number of EBP-2 comets from the cell body in wt, *unc-116* and *gip-1* mutants. The quantification is shown in Author response image 1.
